# FedGC: An Accurate and Efficient Federated Learning under Gradient Constraint for Heterogeneous Data

## Abstract

Federated Learning (FL) is an important paradigm in large-scale distributed machine learning, which enables multiple clients to jointly learn a unified global model without transmitting their local data to a central server. FL has attracted growing attentions in many real-world applications, such as multi-center cardiovascular disease diagnosis and autonomous driving. Practically, the data across clients are always heterogeneous, i.e., not independently and identically distributed (Non-IID), making the local models suffer from catastrophic forgetting of the initial (or global) model. To mitigate this forgetting issue, existing FL methods may require additional regularization terms or generate pseudo data, resulting to 1) limited accuracy; 2) long training time and slow convergence rate for real-time applications; and 3) high communication cost. In this work, an accurate and efficient *Federated Learning algorithm under Gradient Constraints* (FedGC) is proposed, which provides three advantages: i) High accuracy is achieved by the proposed *Client-Gradient-Constraint based projection method* (CGC) to alleviate the forgetting issue occurred in clients, and the proposed *Server-Gradient-Constraint based projection method* (SGC) to effectively aggregate the gradients of clients; ii) Short training time and fast convergence rate are enabled by the proposed fast *Pseudo-gradient-based mini-batch Gradient Descent* (PGD) method and SGC; iii) Low communication cost is required due to the fast convergence rate and only gradients are necessary to be transmitted between server and clients. In the experiments, four real-world image datasets with three Non-IID types are evaluated, and five popular FL methods are used for comparison. The experimental results demonstrate that our FedGC not only significantly improves the accuracy and convergence rate on Non-IID data, but also drastically decreases the training time. Compared to the state-of-art FedReg, our FedGC improves the accuracy by up to 14.28% and speeds up the local training time by 15.5 times while decreasing 23% of the communication cost.

## 1 Introduction

Federated Learning (FL) enables multiple participations / clients to collaboratively train a global model while keeping the training data local due to various concerns such as data privacy and real-time processing. FL has attracted growing attention in many real-world applications, such as multi-center cardiovascular disease diagnosis Linardos et al. (2022), Homomorphic Encryption-based healthcare system Zhang et al. (2022), FL-based real-time autonomous driving Zhang et al. (2021a); Nguyen et al. (2022), FL-based privacy-preserving vehicular navigation Kong et al. (2021), FL-based automatic trajectory prediction Majcherczyk et al. (2021); Wang et al. (2022). However, in practice, the data across clients are always *heterogeneous*, i.e., *not independently and identically distributed* (Non-IID) (Sattler et al., 2020; Zhang et al., 2021b), which hinders the optimization convergence and generalization performance of FL in real-word applications. At each communication round, a client firstly receives the aggregated knowledge of all clients from the server and then locally trains its model using its own data. If the data are Non-IID across clients, the local optimum of each client can be far from the others after local training and the initial model parameters received from server will be overridden. Hence, the clients will forget the initially received knowledge from

the server, i.e., the clients suffer from the *catastrophic forgetting* of the learned knowledge from other clients Shoham et al. (2019); Xu et al. (2022). In other words, there is a drastic performance drop (or loss increase) of model on global data after local training (as detailed in Appendix A.10).

Recently, several approaches have been proposed to mitigate the catastrophic forgetting in FL, e.g., Federated Curvature (FedCurv) (Shoham et al., 2019) and FedReg Xu et al. (2022). FedCurv utilizes the continual learning method Elastic Weight Consolidation (EWC) (Kirkpatrick et al., 2017) to penalize the clients for changing the most informative parameters. The Fisher information matrix is used in EWC to determine which parameters are informative. However, EWC is not effective for mitigating the catastrophic forgetting Xu et al. (2022) in FL, and FedCurv needs to transmit the Fisher matrix between the server and clients besides model parameters. That significantly increases the communication cost (2.5 times than the baseline FedAvg Xu et al. (2022)). In addition, the calculation of Fisher matrix drastically increases the local training time. FedReg (Xu et al., 2022) is the most recently proposed FL method inspired by the continual learning method Gradient Episodic Memory (GEM) Lopez-Paz & Ranzato (2017). GEM alleviates the catastrophic forgetting by avoiding the increase of loss at previous tasks. However, it requires an episodic memory to contain the representative samples from all previous tasks, which hinders it from being suitable for FL due to data privacy concerns Xu et al. (2022). To resolve this, each client in FedReg firstly generates pseudo data by encoding the knowledge of previous training data learned by the global model, and then regularizes its model parameters by avoiding the increase of loss on pseudo data after local training. Although it uses generated pseudo data to protect data privacy and alleviate the forgetting issue in FL, the data generation will increase a lot of computational and storage costs for clients, especially when clients have large-scale data. In addition, the generation of pseudo data and parameter regularization also significantly increase the local training time. Therefore, these methods are not friendly enough to many real-time applications that concern communication & computational costs.

In this work, we propose an accurate and efficient *Federated Learning algorithm under Gradient Constraints* (FedGC) to improve the performance of FL on Non-IID data and reduce the local training time. **At client**, a fast *Pseudo-gradient-based mini-batch Gradient Descent* (PGD) algorithm is proposed to reduce the local training time while accelerating the convergence rates of FL. The pseudo gradient of a local model is obtained by calculating its gradients over few mini-batches data using gradient descent algorithm. In addition, to mitigate catastrophic forgetting, we propose an effective *Client-Gradient-Constraint based projection method* (CGC). Different from GEM requiring memorized data from other clients and FedReg generating pseudo data at clients, our CGC only utilizes the server gradient (i.e., the aggregated gradient from all clients) to restrict the projected gradient to satisfy the constraint: the angle between these two gradients is less than $90°$, in order to enable the local model retains more knowledge received from server. Meanwhile, the projected gradient is also forced to be as close as possible to the pseudo gradient, that enables the local model to learn new knowledge from local data. **At server**, we propose a *Server-Gradient-Constraint based projection method* (SGC) to achieve an optimal server gradient which involves the information of clients participated in aggregation while accelerating the convergence rate by restricting the angles between the server gradient and gradients of participating clients to be less than $90°$. Moreover, our FedGC only transmits the gradients between the server and clients. In other words, our FedGC greatly saves communication costs. The contributions are summarized as follows,

i) High accuracy of our FedGC on Non-IID data is achieved by the proposed CGC to mitigate the catastrophic forgetting occurred in clients and the proposed SGC to effectively aggregate the gradients of clients;

ii) Short training time and fast convergence rate in our FedGC are enabled by the proposed fast PGD method and SGC;

iii) Low communication cost is required in our FedGC due to the fast convergence rate and only gradients to be transmitted between server and clients;

iv) Extensive experimental results illustrate that our FedGC not only improves the performance of FL on Non-IID data with a fast convergence rate but also significantly reduces local training time.

## 2 RELATED WORKS

Federated learning is an important paradigm in large-scale distributed machine learning. It enables multiple clients to jointly learn a unified global model without transmitting their local data to a central server (McMahan et al., 2017; Bhagoji et al., 2019; Yang et al., 2019). FedAvg (McMahan et al., 2017) is the most popular FL algorithm. In FedAvg, clients first locally train models on local data, and then their model updates (e.g., parameters) are transmitted over the network to a central server, where the updates are aggregated. However, data in many real-world applications are always Non-IID, which degrades the performance of FedAvg and slows down the convergence rate Li et al. (2020; 2018); Xu et al. (2022). Many approaches have been proposed to improve the performance and accelerate the convergence rate of FedAvg. Except FedCurv and FedReg, two more popular FL methods resolving the Non-IID are introduced here, including FedProx (Li et al., 2018) and Stochastic Controlled Averaging algorithm (SCAFFOLD) Karimireddy et al. (2020). Similar to FedCurv, FedProx (Li et al., 2018) tackles the heterogeneity in FL by adding a regularization term (i.e., the Proximal term) in local objective function. The proximal term represents the $l_2$ distance between the local model parameters and the initial (global) model parameters. During local training, FedProx minimizes the local loss while restricting the local updates to be close to the initial (global) model. FedProx takes the same communication costs as FedAvg because it does not transmit additional information besides model parameters. However, compared to FedCurv, it lacks of flexibility of model parameters and its stiffness comes at the expense of accuracy. SCAFFOLD (Karimireddy et al., 2020) introduces the control variates $c$ and $c_i$ to correct the client-drift on Non-IID data in local training. The control variates $c$ and $c_i$ aim to guide the local model to update based on the average gradient of participating clients in the previous communication round. However, the variate $c_i$ has the same dimension with the gradient and is transmitted between clients and server, which doubles the communication costs compared with FedAvg. Moreover, the average gradient in the previous communication round may not satisfy its assumptions that $c_j \approx g_j(y_i)$ and $c \approx \frac{1}{N} \sum_j g_j(y_i)$, especially when deep learning models on image datasets perform at clients Li et al. (2021).

Notably, all the above FL methods rely on stochastic gradient descent (SGD) to train models at clients by performing multiple epochs on full local data. That significantly increases the local training time of clients. In addition, the clients owning small training data need to wait a long time before the clients with large-scale data complete local training. That is not computationally efficient in practice and increases the latency between clients. Minibatch SGD (Woodworth et al., 2020) is recently proposed to perform local training of clients on several mini-batch data at the same model, and then it calculates the mean gradient by averaging the gradients.

## 3 METHOD

Given $K$ clients, each client $k$ has a local dataset $\mathcal{D}_k$. $\mathbb{D}_k = \{\boldsymbol{x}_k, \boldsymbol{y}_k\} \subseteq \mathcal{D}_k$ represents a few mini-batches data including $n_k$ samples. Let $T$ be the number of communication rounds, $B$ be the number of mini-batches for local training. At each communication round, a subset of clients $\mathcal{K} \subseteq [K]$ are sampled uniformly like FedAvg. In the $t$-th communication round, $\boldsymbol{\theta}^t$ represents the parameters of the global model at server and $\boldsymbol{\theta}_k^t$ is the parameters of the local model at client $k$. $\boldsymbol{g}_k^t$ represents the gradient of client $k$ on its local data after local training, $\boldsymbol{g}^t$ is the server gradient that is obtained by aggregating local gradients of clients. Notably, the gradient mentioned in this work is the negative gradient for the convenience of calculation.

### 3.1 LOCAL TRAINING WITH PGD AND CGC

At the $t$-th communication round, clients would firstly receive $\boldsymbol{g}^{t-1}$ of the previous round, and then synchronize the parameters of local models with $\boldsymbol{g}^{t-1}$ to ensure clients have the same initially parameters. For client $k \in \mathcal{K}$ with its mini-batches data $\mathbb{D}_k$, the objective function is given by,

$$\min_{\boldsymbol{\theta}_k^t} \ L(f(\boldsymbol{x}_k; \boldsymbol{\theta}_k^t), \boldsymbol{y}_k) \quad \text{s.t.} \quad \left\langle \frac{\partial L(f(\boldsymbol{x}_k; \boldsymbol{\theta}_k^t), \boldsymbol{y}_k)}{\partial \boldsymbol{\theta}_k^t}, \boldsymbol{g}^{t-1} \right\rangle \geq 0 \quad (1)$$

where $\langle . \rangle$ represents the inner product operation. $f(\boldsymbol{\theta}_k^t, \boldsymbol{x}_k)$ represents the prediction of local model at client $k$ with parameters $\boldsymbol{\theta}_k^t$ and input $\boldsymbol{x}_k$, $L(f(\boldsymbol{\theta}_k^t, \boldsymbol{x}_k), \boldsymbol{y}_k)$ is the loss function of client $k$ on its local data $\mathbb{D}_k$ and

$$L(f(\boldsymbol{x}_k; \boldsymbol{\theta}_k^t), \boldsymbol{y}_k) = \frac{1}{|\mathbb{D}_k|} \sum_{(x_{k,i}, y_{k,i} \in \mathbb{D}_k)} L(f(\boldsymbol{x}_{k,i}; \boldsymbol{\theta}_k^t), \boldsymbol{y}_{k,i})) \tag{2}$$

The constraint in problem (1) indicates that the angle between the current gradient $\frac{\partial L(f(\boldsymbol{x}_k; \boldsymbol{\theta}_k^t), \boldsymbol{y}_k)}{\partial \boldsymbol{\theta}_k^t}$ and the server gradient $\boldsymbol{g}^{t-1}$ is less than $90°$. In this way, during local training, the update direction of local model will be not only learned from local data but also restricted by the server gradient. The server gradient obtained by our SGC involves the update direction of clients participated in aggregation (as detailed in Section 3.2). Hence, through solving problem (1), the local model will convergence to the global optimum.

However, (1) is an optimization problem with inequality constraints, which cannot be directly solved by SGD. Hence, we divided the solution into two steps, as shown in Figure 1.

**Step 1: Pseudo-gradient-based mini-batch Gradient Descent (PGD)**

At $t$-th communication round, for client $k \in \mathcal{K}$ with its local data $\mathbb{D}_k$, the loss function Eq. (3) is first minimized,

$$\min_{\boldsymbol{\theta}_{k,B}^t} L(f(\boldsymbol{x}_k; \boldsymbol{\theta}_{k,B}^t), \boldsymbol{y}_k) \tag{3}$$

where $\boldsymbol{\theta}_{k,B}^t$ is the model parameters after $B$ mini-batches.

**Mini-batch update**: Instead of performing multiple epochs on full local data in each communication round like in FedAvg, we prefer to use few mini-batches to train local model to save local training time and avoid the latency between clients. Most importantly, when data are Non-IID, too many iterations in local training may cause local models to be biased towards their local data and enlarge the difference between the local model and the global model. This is not conducive to the convergence of global model. The experiments of FedAvg using mini-batch update in Appendix A.7 can verify this statement.

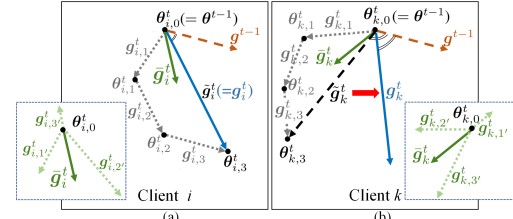

Figure 1: Local training with PGD and CGC performed at two clients of the $t$-th round. Clients receive the server gradient $\boldsymbol{g}^{t-1}$ before local training. At step 1, local models perform PGD on three mini-batches, and compute the pseudo gradients (i.e., $\tilde{\boldsymbol{g}}_i^t$ and $\tilde{\boldsymbol{g}}_k^t$) by (4). At step 2, clients obtain the projected gradients by performing CGC on the pseudo gradients. Since the angle between $\tilde{\boldsymbol{g}}_i^t$ and $\boldsymbol{g}^{t-1}$ is less than $90°$, the projected gradient of client $i$ is $\tilde{\boldsymbol{g}}_i^t$ itself through CGC. In contrast, the angle between $\tilde{\boldsymbol{g}}_k^t$ and $\boldsymbol{g}^{t-1}$ is more than $90°$, the projected gradient $\boldsymbol{g}_k^t$ is obtained by performing CGC (red arrow) on $\tilde{\boldsymbol{g}}_k^t$ to satisfy constraints of (5). $\bar{\boldsymbol{g}}_i^t$ and $\bar{\boldsymbol{g}}_k^t$ (green arrows) are the mean gradients used in Minibatch SGD. The projected gradients usually have larger modulus than the mean gradients.

**Pseudo-gradient**: During this local training, $\boldsymbol{g}_{k,1}^t, \boldsymbol{g}_{k,2}^t, \cdots, \boldsymbol{g}_{k,B}^t$ in Figure 1 (b) are obtained. Different from the mean gradient $\bar{\boldsymbol{g}}_k^t$ $(= \frac{1}{B} \sum_{b=1}^{B} \boldsymbol{g}_{k,b'}^t)$ used in Minibatch SGD that averages multiple gradients for several mini-batches data at the same point (e.g., $\boldsymbol{\theta}_{i,0}^t$), we prefer to calculate a pseudo gradient $\tilde{\boldsymbol{g}}_k^t$ to represent the final gradient after this local training, as shown below.

$$\tilde{\boldsymbol{g}}_k^t = \frac{(\boldsymbol{\theta}_{k,B}^t - \boldsymbol{\theta}_{k,0}^t)}{\eta} \tag{4}$$

where $\boldsymbol{\theta}_{k,0}^t$ is the initialized model parameters and $\eta$ is the learning rate.

Compared to the mean gradient $\bar{\boldsymbol{g}}_k^t$ that has similar modulus with each gradient $\boldsymbol{g}_{k,l}^t, 1 \leq l \leq B$, the pseudo gradient $\tilde{\boldsymbol{g}}_k^t$ would have larger modulus than $\boldsymbol{g}_{k,l}^t$, as shown in Figure 1(b). In this way, $\tilde{\boldsymbol{g}}_k^t$ will promote a large update for local model even with only few mini-batches in local training. This will accelerate the convergence rate of FL.

**Step 2: Client-Gradient-Constraint based projection method (CGC)**

Since the data cross clients are Non-IID, the angle between the pseudo gradient $\tilde{\boldsymbol{g}}_k^t$ and server gradient $\boldsymbol{g}^{t-1}$ may be more than $90°$, as shown in Figure 1(b). That means the update direction of local model deviates from the global optimum, i.e., the catastrophic forgetting occurs at clients. To mitigate this forgetting, the *Client-Gradient-Constraint based projection method* (CGC) is proposed in this subsection.

At $t$-th communication round, for client $k \in \mathcal{K}$ with its local data $\mathbb{D}_k$, the optimization problem of our CGC is given by (5). In problem (5), the projected gradient $\boldsymbol{g}_k^t$ should be as close as possible

to the pseudo gradient $\tilde{g}_k^t$ (in squared L2 norm) while being at an acute angle to the server gradient $g^{t-1}$.

$$\min_{g_k^t} \frac{1}{2} \left\| \tilde{g}_k^t - g_k^t \right\|^2 \quad \text{s.t.} \left\langle g_k^t, g^{t-1} \right\rangle - C \geq 0 \tag{5}$$

where $C = 1e-3$ is a small positive constant to avoid $g_k^t$ and $g^{t-1}$ being orthogonal (i.e., to ensure $\left\langle g_k^t, g^{t-1} \right\rangle > 0$) after projection.

Through solving problem (5), the projected gradient $g_k^t$ will enable the model retain more knowledge received from server while learning new knowledge from local data, i.e., the model can balance preserving the knowledge received from the server and being adaptive to local data.

By simplifying problem (5), we obtain the primal of CGC Quadratic Program (QP) with inequality constraints:

$$\min_u \frac{1}{2} u^T u - h^T u + \frac{1}{2} h^T h \quad \text{s.t.} \ C - z^T u \leq 0 \tag{6}$$

where $\frac{1}{2} h^T h$ is a constant term and can be discarded, $u = g_k^t \in \mathbb{R}^p$, $h = \tilde{g}_k^t \in \mathbb{R}^p$ and $z = g^{t-1} \in \mathbb{R}^p$, and $p$ indicates the number of parameters of local model.

Problem (6) s a QP on $p$ variables and could be measured in millions. Hence, to solve problem (6) efficiently, we convert the primal problem into dual problem, and obtain the dual of the CGC QP:

$$\min_v \frac{1}{2} v^2 z^T z + v \left( h^T z - C \right) \quad \text{s.t.} \ v \geq 0 \tag{7}$$

where $v \in \mathbb{R}$ is a Lagrange multiplier, and problem (7) is a QP on $1 \ll p$ variable. To solve problem (7), the python library *quadprog* [1] is used and then the optimum $v^\star$ is obtained. The details of calculating dual of CGC QP are provided in Appendix A.1.

Finally, the optimal solution to problem (6) is calculated by $u^\star = h + v^\star z$, i.e., $g_k^t = \tilde{g}_k^t + v^\star g^{t-1}$ after our CGC. In addition, since the whole concatenated gradient has very large dimension, we iteratively perform the gradient projection layer by layer (i.e., layer-wise manner) to reduce the memory overhead. Theoretical analysis of gradient projection is detailed in Appendix A.3

## 3.2 SERVER AGGREGATION WITH SGC

At $t$-th communication round, the local gradients (i.e., $g_k^t$) of clients are then send to server for aggregation after local training. At server, the aggregated gradient $g^t$ (i.e., server gradient) is then send back to clients and the parameters $\theta^t$ of global model is calculated by $\theta^t = \theta^{t-1} + \eta g^t$. For aggregation, most FL methods simply use the weighted average of local gradients at server. When the data across clients are Non-IID, the weighted-average may be only effective for few clients. That is because the server gradient may point to the opposite directions of some local gradients (i.e., the angle between them are more than $90°$), that slow down the convergence rate of FL.

To effectively aggregate the local gradients at server and accelerate the convergence rate, we proposed a *Server-Gradient-Constraint based projection method* (SGC). Through our SGC, the projected gradient can point to the positive directions of most local gradients. The optimization problem of SGC is given by,

$$\min_{g^t} \frac{1}{2} \left\| g^t - \bar{g}^t \right\|^2 \quad \text{s.t.} \left\langle g^t, g_k^t \right\rangle - C \geq 0, \ \forall k \in \mathcal{K} \tag{8}$$

where $\bar{g}^t = \sum_{k \in \mathcal{K}} \frac{n_k}{N} g_k^t$ is the weighted average of the local gradients, and $N = \sum_{k \in \mathcal{K}} n_k$. $g_k^t$ is the local gradient of client $k$. The constraint in problem (8) is to restrict the angle between the projected gradient $g^t$ and local gradients at participating clients are less than $90°$.

Similarly, we obtain the primal of SGC QP with inequality constrains by simplifying Eq. (8)

$$\min_z \frac{1}{2} z^T z - g^T z + \frac{1}{2} g^T g \quad \text{s.t.} \ C - Gz \leq 0 \tag{9}$$

where $z = g^t \in \mathbb{R}^p$, $g = \bar{g}^t \in \mathbb{R}^p$, and $G = (\ldots, g_k^t, \ldots)^T \in \mathbb{R}^{|\mathcal{K}| \times p}$, $k \in \mathcal{K}$. $\frac{1}{2} g^T g$ is a constant term and can be ignored. Problem (9) is a QP on $p$ variables. To solve problem (8) efficiently, we also convert the primal problem (9) into dual problem and obtain the dual of the SGC QP:

$$\min_\lambda \left( Gg - C \right)^T \lambda + \frac{1}{2} \lambda^T GG^T \lambda \quad \text{s.t.} \ \lambda \geq 0 \tag{10}$$

where $\lambda \in \mathbb{R}^{|\mathcal{K}|}$ is the Lagrange multiplier, and the problem (10) is a QP on $|\mathcal{K}| \ll p$ variables. The solution $\lambda^\star$ of problem (10) is also obtained by the *quadprog* library. The details of calculating dual of SGC QP are provided in Appendix A.2.

---

[1]https://github.com/quadprog/quadprog

Finally, the optimal solution to problem (9) is calculated by $\boldsymbol{z}^\star = \boldsymbol{g} + \boldsymbol{G}^T\boldsymbol{\lambda}^\star$, i.e., $\boldsymbol{g}^t = \bar{\boldsymbol{g}}^t + \boldsymbol{G}^T\boldsymbol{\lambda}^\star$ after our SGC. When the problem (9) is unsolvable, we simply use $\bar{\boldsymbol{g}}^t$ to be $\boldsymbol{g}^t$.

The pseudo codes of our method are provided in Algorithm 1, and the convergence analysis is detailed in Appendix A.4.

## 4 EXPERIMENTS

We conduct extensive experiments to compare our FedGC with several popular approaches, including FedAvg, FedProx, FedCurv, SCAFFOLD and FedReg, on real image datasets. The data preparation and experimental details are described below. The performances are evaluated in three aspects: 1) Overall test accuracy; 2) Convergence rate and training time; 3) Communication costs.

### 4.1 DATASETS

The experiments are conducted on three real image datasets, including Handwritten-Digits, CIFAR-10 (Krizhevsky, 2009) and CIFAR-100 Krizhevsky (2009). The data preparation of each dataset is described below. More datasets details are provided in Appendix A.6

**Handwritten-Digits** It contains four common handwritten-digits datasets, including MNIST LeCun et al. (1998), MNIST-M Zhao et al. (2022), USPS (Hull, 1994) and SVHN Netzer et al. (2011). Each dataset contains 10 classes. To input these images into deep models sharing the same network architecture, all images in these four datasets are pre-processed by reshaping the size to (32, 32, 3), including the training sets and test sets. The training data are split into 4 clients (named HWDigits-4) and 40 clients (named HWDigits-40) respectively. In HWDigits-4, each client owns one handwritten-digits dataset. In HWDigits-40, each client has images belonging to only one class in a dataset under the one-class setting. HWDigits-4 suffers from the attribute skew of Non-IID issue, in which the attributes (i.e., data features) across clients are different. In HWDigits-40, the data across clients may have different labels and attributes, and hence it suffers from both attribute and label skew.

---

**Algorithm 1: FedGC**

**Input:** $K, T, B$, datasets $\mathcal{D} = \cup_{k \in [K]} \mathcal{D}_k, \eta$.
**Output:** the parameters $\boldsymbol{\theta}^T$ of the global model.

1 Initialize: server $\boldsymbol{\theta}^0$; clients $\boldsymbol{\theta}_k^0 \leftarrow \boldsymbol{\theta}^0$, for $k \in [K]$;
2 **for** $t = 1$ to $T$ **do**
3     **for** $k \in [K]$ *in parallel* **do**
4         `# Synchronize parameters`
5         **if** $t > 1$ **then**
6             $\boldsymbol{g}_k^t \leftarrow \boldsymbol{g}^{t-1}$;
7             $\boldsymbol{\theta}_k^t \leftarrow \boldsymbol{\theta}_k^{t-1} + \eta \boldsymbol{g}_k^t$;
8     **end**
9     Randomly sample clients $\mathcal{K} \subseteq [K]$;
10    **for** $k \in \mathcal{K}$ *in parallel* **do**
11        **for** $b = 1$ to $B$ **do**
12           Randomly sample a mini-batch data $\mathcal{B}_{k,b} \subset \mathcal{D}_k$;
13           $\boldsymbol{\theta}_{k,b}^t \leftarrow SGD\,(\boldsymbol{\theta}_{k,b-1}^t {}^a, \eta)$ on $\mathcal{B}_{k,b}$;
14        **end**
15        $\tilde{\boldsymbol{g}}_k^t = \frac{\boldsymbol{\theta}_{k,B}^t - \boldsymbol{\theta}_{k,0}^t}{\eta}$; `# Pseudo gradient`
16        $\boldsymbol{g}_k^t \leftarrow$ **CGC** $(\tilde{\boldsymbol{g}}_k^t, \boldsymbol{g}^{t-1})$ in (6);
17        $\boldsymbol{\theta}_k^t \leftarrow \boldsymbol{\theta}_{k,0}^t$;
18    **end**
19    $\bar{\boldsymbol{g}}^t \leftarrow \sum_{k \in \mathcal{K}} \frac{n_k}{N} \boldsymbol{g}_k^t$; $\boldsymbol{G} \leftarrow (\ldots, \boldsymbol{g}_k^t, \ldots)^T$, $k \in \mathcal{K}$;
20    $\boldsymbol{g}^t \leftarrow$ **SGC** $(\bar{\boldsymbol{g}}^t, \boldsymbol{G})$ in (9);
21    $\boldsymbol{\theta}^t \leftarrow \boldsymbol{\theta}^{t-1} + \eta \boldsymbol{g}^t$;
22 **end**

---

[a] If $t = 1, \boldsymbol{\theta}_{k,0}^t = \boldsymbol{\theta}_k^{t-1}$, otherwise $\boldsymbol{\theta}_{k,0}^t = \boldsymbol{\theta}_k^t$

**CIFAR-10** and **CIFAR-100** Under the one-class setting, the training sets in CIFAR-10 and CIFAR-100 are split into 10 (named CIFAR10-10) and 100 (named CIFAR100-100) clients, respectively, i.e., each client owns only samples of one class. All clients share the test sets in original CIFAR-10 and CIFAR-100 respectively. CIFAR10-10 and CIFAR100-100 both suffer from the label skew under the one-class setting.

### 4.2 EXPERIMENTAL SETTING

We implement all methods in PyTorch-1.9.0 (Paszke et al., 2019) on a Ubuntu server with two Intel Xen CPUs and 8 NVIDIA RTX 3090 GPUs. SGD with weight decay 0 and momentum 0 is used to train local models at clients. For HWDigits-4 and HWDigits-40, a CNN similar to (McMahan et al., 2017) is adopted in experiments. It contains two 5×5 convolution layers followed by 2×2 Max-Pooling and two fully connected layers (the first layer with 512 units and the second layer with 100 units) with ReLU activation. The communication round $T = 600$ and the number of local epochs $E$ in other compared methods is set to 1. The number of local iterations $B$ (i.e., the number of mini-

batches for local training) in FedGC is 50. The client fraction is set to 1.0 and the batch size is 100. The learning rates $\eta$ of all methods in experiments are tuned in the range of $\{0.01, 0.05, 0.1, 1.0\}$, and the optimal $\eta$ of our FedGC is 0.1 and the ones for all other compared methods are 0.01. The learning rate $\eta$ in FedGC is set to 0.1 and the learning rates in other methods are 0.01. For CIFAR10-10 and CIFAR100-100, a ResNet-9 network with the Fixup initialization (Xu et al., 2022) is trained from scratch. The communication rounds $T$ is set to 500 for CIFAR10-10 and 1000 for CIFAR100-100. $E = 1$ and $B = 1$. The optimal $\eta$ for our FedGC is 0.1 and the ones for all other compared methods are 0.05.

## 4.3 OVERALL TEST ACCURACY

Table 1: Comparison results (%) on four datasets (↑). 'w/o' and 'w' denote the server aggregation in FedGC without and with SGC, respectively.

| Type of Non-IID | Attribute skew | Label skew | | Attribute&Label skew |
|---|---|---|---|---|
| Method | HWDigits-4 | CIFAR10-10 | CIFAR100-100 | HWDigits-40 |
| FedAvg | 93.07 | 55.34 | 40.59 | 62.11 |
| FedProx | 93.69 | 55.13 | 32.74 | 62.01 |
| FedCurv | 91.78 | 55.61 | 9.48[1] | 62.55 |
| SCAFFOLD | 92.29 | 34.40 | 2.62[1] | 45.32 |
| FedReg | 92.32 | 57.70 | 53.02 | 63.05 |
| Our FedGC(w/o) | *94.15* | *59.64* | *55.73* | *74.63* |
| Our FedGC(w) | **95.27** | **60.92** | **56.56** | **77.33** |
| #samples per client | 7,291∼73,257 | 5,000 | 500 | 542∼13,861 |

[1] It represents model fails to convergence after 1000 communication rounds.

Table 1 shows the overall test accuracies of all compared methods on HWDigits-4, HWDigits-40, CIFAR10-10 and CIFAR100-100. In Table 1, our FedGC (denotes FedGC(w) if not specified) achieves the highest test accuracies than all other compared methods. Specifically, for HWDigits-40, our FedGC outperforms the SOTA FedReg by 14.28% (=77.33-63.05) of accuracy, which clearly shows the superiority of our FedGC in tackling the Non-IID data. FedCurv is designed based on the assumption that the deep neural networks are over-parameterized enough, so that it has a good probability of finding an optimal solution to task B in the neighborhood of previously learned task solution. However, for CIFAR100-100, there are 100 clients and each client contains only samples of one class. Hence, the assumption of over-parameterization cannot be satisfied and it only gets 9.48% of accuracy. FedProx has strong constraints on model parameters, so that it lacks of flexibility, which hinders the learning of new knowledge from the local data. Hence, on CIFAR100-100, it also gets worse performance than the baseline FedAvg. SCAFFOLD gets the worst performances on most of the compared datasets, because the average gradient in the previous communication round may not satisfy its assumptions, especially when deep learning models on image datasets perform at clients Li et al. (2021). FedReg is the most recently proposed method, which alleviates the catastrophic forgetting in FL by generating pseudo data. The pseudo data is generated to guarantee that the loss of local model on them is less than that of initial (global) model on them. In this way, the catastrophic forgetting issues can be alleviated. However, when the number of samples between clients varies significantly (e.g., HWDigits-4 and HWDigits-40), the FedReg will be biased to the majority clients (i.e., the clients with many samples). That is because the global model is obtained by weighted averaging the model parameters in FedReg and the global model will be biased to majority clients. In local training, the client with few samples (i.e., minority clients) will forget the learned knowledge from local data after regularizing the model parameters with pseudo data in FedReg. In other words, the FedReg has further magnified the bias of global model to majority clients.

Benefited from the proposed CGC, our FedGC mitigates the catastrophic forgetting of FL by constraining the local gradient at an acute angle to the server gradient and simultaneously minimizing the loss of local model on its local data. Moreover, benefited from the proposed SGC, the projected server gradient in our FedGC can effectively aggregate the knowledge from clients. The gradient projected by both CGC and SGC can reduce the bias toward majority clients. As shown in Table 1, FedGC(w) further improves the performances of FedGC(w/o). Particularly, the SGC can further accelerate the convergence rate of FL (as detailed in Section 4.4). Hence, FedGC(w) achieves the highest accuracy on all compared datasets, and the improvements are up to 14.28% (HWDigits-40) compared to the state-of-the-art (SOTA) FedReg. In addition, our FedGC neither transmits extra

data across the server and clients, nor needs extra storage costs to keep the generated data. The computational costs of CGC and SGC are also extremely small. Therefore, FedGC is very friendly for edge devices with limited computational resources.

Figure 2: Test accuracy vs. communication rounds on four datasets. The black dotted line denotes the highest accuracy of FedReg. The red and blue dotted lines represent the round numbers of FedGC and FedReg reaching this accuracy, respectively.

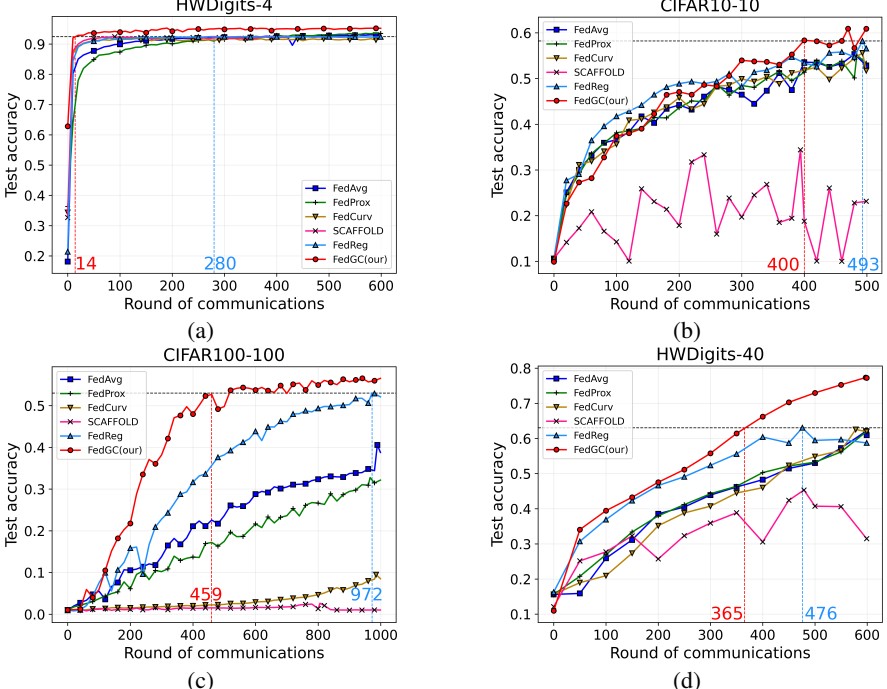

(a)  (b)

(c)  (d)

## 4.4 CONVERGENCE RATE AND TRAINING TIME

Figure 2 illustrates that our FedGC can quickly achieve higher accuracies than other methods on all compared datasets until the end of communication. It indicates that our method can improve both the performance and the convergence rate of FL on Non-IID data. The high convergence rate is mainly benefited from our proposed PGD and SGC. FedReg is also proposed for accelerating the convergence rate of FL by alleviating the forgetting issue, but it is more biased to majority clients, that may slow down its convergence rate on HWDigits-4 and HWDigits-40. For HWDigits-40, FedReg achieves its highest accuracy (63.05%) at round 476 as shown in Figure 2(d) while our FedGC reaches this accuracy at round 365. Meanwhile, our FedGC improves the accuracy by 14.28% while decreasing 23% ($\approx 1 - \frac{365 \times |\boldsymbol{\theta}|}{476 \times |\boldsymbol{\theta}|}$) of communication cost[2] compared to FedReg. For CIFAR100-100, our FedGC also quickly reaches 53.02% of accuracy (i.e., the accuracy of FedReg), which empirically verifies that the projected gradients in both server and clients are helpful for FL. Among compared methods, SCAFFOLD is very unstable during training (as shown in Figures 2 (b) and (d)). In addition, Figure 3 illustrates that our SGC can further improve the convergence rate of FL.

Training time is another effective way to measure the practicality and efficiency of FL methods. The average training time of each client per communication round are summarized in Table 2, where FedAvg is considered as the baseline. Due to the calculation of proximal term in objective function during local training, FedProx increases training time by about 13s [3] at each dataset per round. Similarly, SCAFFOLD also requires more time to calculate and transmit the control variate between clients and server. FedCurv dramatically increases the training time due to the time consuming calculation of Fisher information matrix in every mini-batch. FedReg needs to generate pseudo data during local training, so that it also significantly increases the training time. On the contrary, in our FedGC, the gradient projection methods in both clients and server are very efficient due to few

---

[2]The calculation is provided in Section 4.5 with FedReg as baseline. $|\boldsymbol{\theta}|$ denotes the amount of parameters.

[3]$\approx \frac{(67.99+65.15+64.65+82.41)-(58.63+44+44.74+78.93)}{4}$

Figure 3: Test accuracy vs. communication rounds on HWDigits-4 and CIFAR10-10. The black dotted line denotes the highest accuracy of FedReg. The blue, green and red dotted lines represent the round numbers of FedReg, FedGC(w/o) and FedGC(w) reaching this accuracy, respectively.

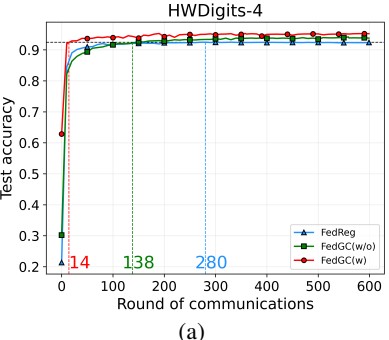

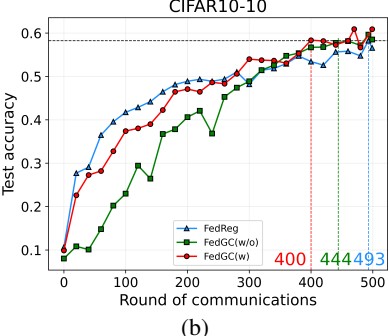

(a)  (b)

variables in the dual problems. Moreover, all compared methods train local model by performing multiple epochs on full local data, while our FedGC trains local model on few mini-batches in each communication round. In this way, it will greatly decrease the training time per round, especially for the clients with many samples (e.g., HWDigits-4, HWDigits-40 and CIFAR100-100). Particularly, our FedGC speeds up the training time by 15.5 ($\approx \frac{274.72}{17.77}$) times compared to SOTA FedReg on HWDigits-40.

Table 2: The training time per communication round and the communication costs (Comm. cost) ($\downarrow$).

| Method | HWDigits-4 | | CIFAR10-10 | | CIFAR100-100 | | HWDigits-40 | |
|---|---|---|---|---|---|---|---|---|
| | Time(s) | Comm. cost | Time(s) | Comm. cost | Time(s) | Comm. cost | Time(s) | Comm. cost |
| FedAvg | 58.63 | $557\times |\boldsymbol{\theta}|$ | 44.00 | $480\times |\boldsymbol{\theta}|$ | 44.74 | $989\times |\boldsymbol{\theta}|$ | 78.93 | $598\times |\boldsymbol{\theta}|$ |
| FedProx | 67.99 | $439\times |\boldsymbol{\theta}|$ | 65.15 | *N.A.* | 64.65 | *N.A.* | 82.41 | *N.A.* |
| FedCurv | 970.42 | *N.A.*[1] | 1055.01 | $490\times 2.5|\boldsymbol{\theta}|$ | 1061.29 | *N.A.* | 933.11 | $578\times 2.5|\boldsymbol{\theta}|$ |
| SCAFFOLD | 81.21 | *N.A.* | 63.99 | *N.A.* | 72.54 | *N.A.* | 115.84 | *N.A.* |
| FedReg | 260.97 | *N.A.* | 257.69 | $431\times |\boldsymbol{\theta}|$ | 260.72 | $531\times |\boldsymbol{\theta}|$ | 274.72 | $476\times |\boldsymbol{\theta}|$ |
| FedGC (our) | **6.33** | $\mathbf{19\times |\boldsymbol{\theta}|}$ | **18.82** | $\mathbf{327\times |\boldsymbol{\theta}|}$ | **39.71** | $\mathbf{287\times |\boldsymbol{\theta}|}$ | **17.77** | $\mathbf{357\times |\boldsymbol{\theta}|}$ |

[1] *N.A.* represents the method has not achieved the target accuracy within maximum communication rounds.

## 4.5 COMMUNICATION COSTS

The comparison of communication costs is shown in Table 2. We calculate the communication cost by the number of communication round when the method achieves the target accuracy (e.g., the highest accuracy of baseline FedAvg) multiplied by the amount of transmitted data (e.g., the amount of parameters $|\boldsymbol{\theta}|$). $2.5|\boldsymbol{\theta}|$ represents that FedCurv needs to transmit the Fisher matrix between the server and clients besides model parameters. In Table 2, our FedGC requires the lowest communication costs on four datasets when achieving the target accuracy. Specifically, on HWDigits-4, our FedGC can decrease 96.6%($\approx 1 - \frac{19\times|\boldsymbol{\theta}|}{557\times|\boldsymbol{\theta}|}$) of the communication cost compared to FedAvg due to the fewest number of communication rounds. Compared to SOTA FedReg, the communication cost can be decreased by 45%($\approx 1 - \frac{287\times|\boldsymbol{\theta}|}{531\times|\boldsymbol{\theta}|}$) on CIFAR100-100.

## 5 CONCLUSIONS

This paper proposes an accurate and efficient Federated Learning algorithm under Gradient Constraints (FedGC) to address the challenging FL scenarios on Non-IID data: Low accuracy, and long local training time. Our FedGC includes several novel modules to tackle the issues of Non-IID data: 1) CGC (alleviating the forgetting issue for higher accuracy) and PGD (for significantly shorter training time and fast convergence rate) at client; 2) SGC (for effective aggregation of local gradients) at server. Moreover, FedGC enables low communication cost due to fast convergence rate and only transmissions of gradients between server and clients. Extensive experiments are conducted on four datasets comparing several SOTA FL methods. The experimental results show that FedGC can significantly improve the performance and convergence rate of FL on Non-IID data with low communication and storage costs. The local training time can be drastically reduced. Our FedGC improves the accuracy by up to 14.28% and speeds up the local training time by 15.5 times compared to the SOTA FedReg, while decreasing 23% of the communication cost. In a nutshell, FedGC has great potential for many real-world applications that concerns performance, real-time, communication & computational costs, and privacy preservation.

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

# A APPENDIX

## A.1 THE DUAL OF CGC QP

Here we provide the procedure of how to convert the primal of CGC QP into the dual problem. The primal problem is:

$$
\begin{aligned}
&\min_{\boldsymbol{u}} \ \frac{1}{2}\boldsymbol{u}^T\boldsymbol{u} - \boldsymbol{h}^T\boldsymbol{u} + \frac{1}{2}\boldsymbol{h}^T\boldsymbol{h} \\
&\text{s.t.} \quad C - \boldsymbol{z}^T\boldsymbol{u} \leq 0
\end{aligned}
\tag{11}
$$

where $\boldsymbol{u} \in \mathbb{R}^p$, $\boldsymbol{h} \in \mathbb{R}^p$ and $\boldsymbol{z} \in \mathbb{R}^p$, and $p$ indicates the number of parameters of local model.

To obtain the dual of (11), we firstly conduct the Lagrange function $L(\boldsymbol{u}, v)$ with a Lagrange multiplier $v \in \mathbb{R}$ as follow

$$
\begin{aligned}
L(\boldsymbol{u}, v) &= \frac{1}{2}\boldsymbol{u}^T\boldsymbol{u} - \boldsymbol{h}^T\boldsymbol{u} + \frac{1}{2}\boldsymbol{h}^T\boldsymbol{h} + v(C - \boldsymbol{z}^T\boldsymbol{u}) \\
&= \frac{1}{2}\boldsymbol{u}^T\boldsymbol{u} - \boldsymbol{h}^T\boldsymbol{u} + \frac{1}{2}\boldsymbol{h}^T\boldsymbol{h} + vC - v\boldsymbol{z}^T\boldsymbol{u}) \\
&= \frac{1}{2}\boldsymbol{u}^T\boldsymbol{u} - (\boldsymbol{h}^T + v\boldsymbol{z}^T)\boldsymbol{u} + \frac{1}{2}\boldsymbol{h}^T\boldsymbol{h} + vC
\end{aligned}
\tag{12}
$$

Because (12) is a quadratic convex function, its minimal value can be obtained by

$$
\nabla_{\boldsymbol{u}} L(\boldsymbol{u}, v) = \boldsymbol{u} - (\boldsymbol{h} + v\boldsymbol{z}) = 0 \tag{13}
$$

$$
\implies \boldsymbol{u} = \boldsymbol{h} + v\boldsymbol{z} \tag{14}
$$

Hence, the minimal value of (12) is

$$
\begin{aligned}
L(\boldsymbol{h} + v\boldsymbol{z}, v) &= \frac{1}{2}(\boldsymbol{h} + v\boldsymbol{z})^T(\boldsymbol{h} + v\boldsymbol{z}) - \boldsymbol{h}^T(\boldsymbol{h} + v\boldsymbol{z}) + \frac{1}{2}\boldsymbol{h}^T\boldsymbol{h} + v(C - \boldsymbol{z}^T(\boldsymbol{h} + v\boldsymbol{z})) \\
&= \frac{1}{2}(\boldsymbol{h}^T + v\boldsymbol{z}^T)(\boldsymbol{h} + v\boldsymbol{z}) - \boldsymbol{h}^T(\boldsymbol{h} + v\boldsymbol{z}) + \frac{1}{2}\boldsymbol{h}^T\boldsymbol{h} + vC - v\boldsymbol{z}^T(\boldsymbol{h} + v\boldsymbol{z}) \\
&= -\frac{1}{2}v^2\boldsymbol{z}^T\boldsymbol{z} - v\boldsymbol{h}^T\boldsymbol{z} + vC
\end{aligned}
\tag{15}
$$

The dual function $G(v)$ is

$$
G(v) = \inf_{\boldsymbol{u}} L(\boldsymbol{u}, v) \tag{16}
$$

and the dual problem is

$$
\begin{aligned}
&\mathbf{max}_v \ \ G(v) \\
&\mathbf{s.t.} \ \ \ v \geq 0
\end{aligned}
\tag{17}
$$

Therefore, the dual of CGC QP is

$$
\begin{aligned}
&\mathbf{min}_v \ \ \frac{1}{2}v^2\boldsymbol{z}^T\boldsymbol{z} + v(\boldsymbol{h}^T\boldsymbol{z} - C) \\
&\mathbf{s.t.} \ \ \ v \geq 0
\end{aligned}
\tag{18}
$$

Problem (18) can be solved by *quadprog* library and the optimum $v^\star$ is obtained. The standard form of QP problem in *quadprog* is

$$
\begin{aligned}
&\mathbf{min}_{\mathbf{x}} \ \ \frac{1}{2}\mathbf{x}^T\mathbf{G}\mathbf{x} - \mathbf{a}^T\mathbf{x} \\
&\mathbf{s.t.} \ \ \mathbf{C}^T\mathbf{x} \leq \mathbf{b}
\end{aligned}
\tag{19}
$$

At last, the optimal solution of the primal CGC OP is calculated by $\boldsymbol{u}^\star = \boldsymbol{h} + v^\star\boldsymbol{z}$.

### A.2 THE DUAL OF SGC QP

Here we provide the procedure of how to convert the primal of SGC QP into the dual problem. The primal problem is:

$$
\begin{aligned}
&\mathbf{min}_{\boldsymbol{z}} \ \ \frac{1}{2}\boldsymbol{z}^T\boldsymbol{z} - \boldsymbol{g}^T\boldsymbol{z} + \frac{1}{2}\boldsymbol{g}^T\boldsymbol{g} \\
&\mathbf{s.t.} \ \ C - \boldsymbol{G}\boldsymbol{z} \leq 0
\end{aligned}
\tag{20}
$$

where $\boldsymbol{z} \in \mathbb{R}^p$, $\boldsymbol{g} \in \mathbb{R}^p$, and $\boldsymbol{G} \in \mathbb{R}^{|\mathcal{K}| \times p}$. Problem (20) is a QP on $p$ variables.

Its Lagrange function with a Lagrange multiplier $\boldsymbol{\lambda} \in \mathbb{R}^{|\mathcal{K}|}$ is

$$L(\boldsymbol{z}, \boldsymbol{\lambda}) = \frac{1}{2}\boldsymbol{z}^T\boldsymbol{z} - \boldsymbol{g}^T\boldsymbol{z} + \frac{1}{2}\boldsymbol{g}^T\boldsymbol{g} + \boldsymbol{\lambda}^T(\boldsymbol{C} - \boldsymbol{G}\boldsymbol{z})$$
$$= \frac{1}{2}\boldsymbol{z}^T\boldsymbol{z} - (\boldsymbol{g}^T + \boldsymbol{\lambda}^T\boldsymbol{G})\boldsymbol{z} + \frac{1}{2}\boldsymbol{g}^T\boldsymbol{g} + \boldsymbol{\lambda}^T\boldsymbol{C} \tag{21}$$

Since $L(\boldsymbol{z}, \boldsymbol{\lambda})$ is a quadratic convex function, its minimal value can be obtained by

$$\nabla_{\boldsymbol{z}}L(\boldsymbol{z}, \boldsymbol{\lambda}) = \boldsymbol{z} - (\boldsymbol{g}^T + \boldsymbol{\lambda}^T\boldsymbol{G}) = 0$$
$$\implies \boldsymbol{z} = \boldsymbol{g} + \boldsymbol{G}^T\boldsymbol{\lambda} \tag{22}$$

So

$$L(\boldsymbol{g} + \boldsymbol{G}^T\boldsymbol{\lambda}, \boldsymbol{\lambda}) = \frac{1}{2}(\boldsymbol{g} + \boldsymbol{G}^T\boldsymbol{\lambda})^T(\boldsymbol{g} + \boldsymbol{G}^T\boldsymbol{\lambda}) - \boldsymbol{g}^T(\boldsymbol{g} + \boldsymbol{G}^T\boldsymbol{\lambda}) + \frac{1}{2}\boldsymbol{g}^T\boldsymbol{g} + \boldsymbol{\lambda}^T\boldsymbol{C} - \boldsymbol{\lambda}^T\boldsymbol{G}(\boldsymbol{g} + \boldsymbol{G}^T\boldsymbol{\lambda})$$
$$= -\frac{1}{2}\boldsymbol{\lambda}^T\boldsymbol{G}\boldsymbol{G}^T\boldsymbol{\lambda} - \frac{1}{2}\boldsymbol{g}^T\boldsymbol{G}^T\boldsymbol{\lambda} - \frac{1}{2}\boldsymbol{\lambda}^T\boldsymbol{G}\boldsymbol{g} + \boldsymbol{\lambda}^T\boldsymbol{C}$$
$$= -\frac{1}{2}\boldsymbol{\lambda}^T\boldsymbol{G}\boldsymbol{G}^T\boldsymbol{\lambda} - \boldsymbol{g}^T\boldsymbol{G}^T\boldsymbol{\lambda} + \boldsymbol{\lambda}^T\boldsymbol{C}$$
$$= -(\boldsymbol{G}\boldsymbol{g} - \boldsymbol{C})^T\boldsymbol{\lambda} - \frac{1}{2}\boldsymbol{\lambda}^T\boldsymbol{G}\boldsymbol{G}^T\boldsymbol{\lambda} \tag{23}$$

Therefore, the dual of SGC QP is

$$\min_{\boldsymbol{\lambda}} \ (\boldsymbol{G}\boldsymbol{g} - \boldsymbol{C})^T\boldsymbol{\lambda} + \frac{1}{2}\boldsymbol{\lambda}^T\boldsymbol{G}\boldsymbol{G}^T\boldsymbol{\lambda}$$
$$\text{s.t.} \ \boldsymbol{\lambda} \geq 0 \tag{24}$$

The optimum $\boldsymbol{\lambda}^\star$ can be calculated by library $quadprog$ and the optimal solution of SGC QP is $\boldsymbol{z}^\star = \boldsymbol{g} + \boldsymbol{G}^T\boldsymbol{\lambda}^\star$.

### A.3 THEORETICAL ANALYSIS OF GRADIENT PROJECTION

**Assumption 1** *In each local update, small optimization steps happen and thus we can assume that the function $F$ is locally linear (i.e., convex function).*

To enable the local model retains more knowledge received from server after local update, the loss of local model on global data $\mathcal{D}$ should reduce after local update, i.e.,

$$F(\boldsymbol{\theta}_k^t, \mathcal{D}) < F(\boldsymbol{\theta}^{t-1}, \mathcal{D}) \tag{25}$$

where $\boldsymbol{\theta}_k^t$ is the model parameters at client $k$ after local update and $\boldsymbol{\theta}^{t-1}$ is the initial model (i.e., the global model at previous round) parameters.

Since the global data $\mathcal{D}$ cannot be achieved in FL, we use gradient projection to achieve this objective. For simplifying, $\mathcal{D}$ is omitted in the following equations.

From convexity we know that $\nabla F\left(\boldsymbol{\theta}^{t-1}\right)^{\mathrm{T}}\left(\boldsymbol{\theta}_k^t - \boldsymbol{\theta}^{t-1}\right) \geq 0$ implies $F(\boldsymbol{\theta}_k^t) \geq F(\boldsymbol{\theta}^{t-1})$, so the search direction in local training must satisfy

$$\nabla F\left(\boldsymbol{\theta}^{t-1}\right)^{\mathrm{T}}\left(\boldsymbol{\theta}_k^t - \boldsymbol{\theta}^{t-1}\right) < 0 \tag{26}$$

Thus, it must make an acute angle with negative gradient, i.e.,

$$\left(\boldsymbol{g}^{t-1}\right)^{\mathrm{T}}\boldsymbol{g}_k^t > 0 \Leftrightarrow \langle \boldsymbol{g}^{t-1}, \boldsymbol{g}_k^t \rangle > 0 \tag{27}$$

where $\boldsymbol{g}^{t-1} = -\nabla F_k\left(\boldsymbol{\theta}^{t-1}\right)$ and $\boldsymbol{g}_k^t = \boldsymbol{\theta}_k^t - \boldsymbol{\theta}^{t-1}$.

### A.4 Convergence analysis

Here we give a simple proof that our FedGC has a faster convergence rate than FedAvg. We analyze the convergence of our FedGC by finding an upper bound $\xi$ of $\mathbb{E}\left[F(\boldsymbol{\theta}^T)\right] - F^\star$, i.e., $\mathbb{E}\left[F(\boldsymbol{\theta}^T)\right] - F^\star \leq \xi$, where $F(\boldsymbol{\theta}^T)$ represents the final global model with parameter $\boldsymbol{\theta}^T$ in FL and $F^\star$ is the optimal model for all clients' data (i.e., the upper bound of global model in FL).

From A.3, our FedGC can guarantee that the loss of local model on global data reduces after local update (i.e., $F(\boldsymbol{\theta}_k^t) < F(\boldsymbol{\theta}^{t-1})$), while FedAvg cannot. In other words, our FedGC can guarantee that the loss of global model gradually reduces (i.e., $F(\boldsymbol{\theta}^t) < F(\boldsymbol{\theta}^{t-1})$), while FedAvg cannot. In detail,

$$F(\boldsymbol{\theta}^t) = F\left(\sum_{k=1}^K p_k \boldsymbol{\theta}_k^t\right) \tag{28}$$

where the aggregation is averaged for simplifying, and $\sum_{k=1}^K p_k = 1$.

Let Assumption 1 holds, we can obtain

$$F(\boldsymbol{\theta}^t) = F\left(\sum_{k=1}^K p_k \boldsymbol{\theta}_k^t\right) = \sum_{k=1}^K p_k F\left(\boldsymbol{\theta}_k^t\right) < \sum_{k=1}^K p_k F\left(\boldsymbol{\theta}^{t-1}\right) = F(\boldsymbol{\theta}^{t-1}) \tag{29}$$

Thus, after $T$ communication rounds (i.e., one fixed $T$), the difference between $\mathbb{E}\left[F(\boldsymbol{\theta}^T)\right]$ and $F^\star$ of our FedGC should be smaller than that of FedAvg. Thus, we can conclude that $\xi^{GC} < \xi^{Avg}$.

### A.5 Privacy-preserving analysis

Recently, Deep Leakage from Gradients (DLG) Zhu et al. (2019) attack has attracted growing attentions, which can completely steal real data from gradients. However, under our FedGC, the real data cannot be reversely obtained. The details are shown below.

In DLG, a pair of "dummy" input and label are first randomly generated to perform the usual forward and backward operations. In order to obtain the real data reversely, the dummy gradients from the dummy data are firstly derived, then DLG optimizes the dummy inputs and labels by minimizing the Euclidean distance between the dummy gradients and the real gradients. **However, matching the gradients cannot make the dummy data close to the real data when our FedGC performs.**

Figure 4: DLG in our FedGC.

In Figure 4, at $t$-th communication round, an honest client $k$ samples a minibatch $(x_{t,k}, y_{t,k})$ from its own data and an evil client randomly initializes a dummy input $x'_{t,k}$ with dummy label $y'_{t,k}$. The

objective of the evil client is

$$x'_{t,k}, y'_{t,k} = \arg \min_{x'_{t,k}, y'_{t,k}} \left\| \ddot{\boldsymbol{g}}_k^t - \boldsymbol{g}_k^t \right\|^2 \tag{30}$$

where

$$\ddot{\boldsymbol{g}}_k^t = \frac{\partial L'\left(F\left(x'_{t,k}, \boldsymbol{\theta}_k^t\right), y'_{t,k}\right)}{\partial \boldsymbol{\theta}_k^t} \tag{31}$$

$$\boldsymbol{g}_k^t = CGC\left(\frac{\partial L\left(F\left(x_{t,k}, \boldsymbol{\theta}_k^t\right), y_{t,k}\right)}{\partial \boldsymbol{\theta}_k^t}, \boldsymbol{g}^{t-1}\right) \tag{32}$$

From Eq.(32), we can observe that in our FedGC, $\boldsymbol{g}_k^t$ is not only determined by $\frac{\partial L}{\partial \boldsymbol{\theta}_k^t}$ but also dependent on the global gradient $\boldsymbol{g}^{t-1}$ by our CGC, and the CGC projection cannot be reversely derived. Thus, we can obtain

$$(x'_{t,k}, x'_{t,k}) \neq (x_{t,k}, y_{t,k}) \tag{33}$$

Therefore, we can conclude that real data cannot be recovered with gradient inversion attacks (e.g., DLG) in our FedGC and our FedGC can ensure preserving of privacy.

In addition, we conduct the gradient inversion attacks experiment to compare FedAvg and our FedGC with DLG on CIFAR10-10 dataset. The backbone is ResNet-9. We set the number of iterations in DLG to 300. As shown in Figure 5, the quality of the images recovered from our FedGC is significantly worse than those recovered from FedAvg, exhibiting a better privacy protection capability of our FedGC.

Figure 5: Images recovered from updated gradients of FedAvg and FedGC on CIFAR10-10.

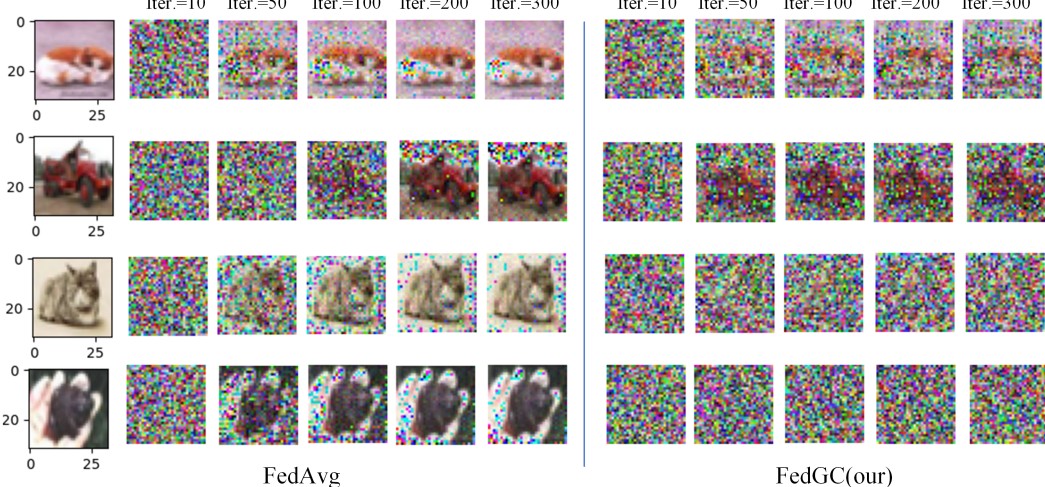

## A.6 More datasets details

**Handwritten-Digits** MNIST has a training sets of 60,000 examples and a test set of 10,000 examples. The images in MNIST are grayscale with image size 28×28. MNIST-M [4] contains 59,001 training and 90,001 test RGB images with size 28×28. USPS contains 9,298 grayscale images with image size 16×16. There are 73,257 RGB images in training set of SVHN [5] and 26,032 RGB images in test set with image size 32×32.

**CIFAR-10** and **CIFAR-100** CIFAR-10 contains 60,000 32×32 color images of 10 classes, including 5,000 images in training set and 1,000 images in test set per class. CIFAR-100 consists of 60,000

---

[4]http://yaroslav.ganin.net/

[5]http://ufldl.stanford.edu/housenumbers/

$32\times32$ color images of 100 classes. In CIFAR-100, there are 500 images in training set and 100 images in test set per class.

## A.7 EXPERIMENT OF FEDAVG ON HWDIGITS-4 UNDER MINI-BATCH UPDATE

In this experiment, the number of mini-batches $B$ for local training in FedAvg is in the range of $\{1, 10, 30, 50\}$, and the number of communication rounds $T$ is 600 for all experiments.

Table 3: Experiment of FedAvg on HWDigits-4 under mini-batch update.

| Setting | $B = 1$ | $B = 10$ | $B = 20$ | $B = 50$ | $E = 1$ (i.e., $B = 73 \sim 732$) |
|---------|---------|----------|----------|----------|-----------------------------------|
| FedAvg  | 85.22   | 92.75    | 93.43    | **93.59** | 93.07                             |

In Table 3, we can observe that FedAvg can achieve better performance when $B = 50$ while requiring less iterations in local training stage compared to $E = 1$. The experimental results are consistent with the statement in Section 3.1.

## A.8 LEARNING CURVES

The learning curves are drawn in Figure 6, where the x-axis is scaled by communication-budget (i.e., in multiples of $|\theta|$). The figures show that our FedGC can reach the target accuracy (i.e., the highest accuracy of baseline FedAvg) with the smallest communication budget.

Figure 6: Learning curves of compared methods in terms of communication budget.

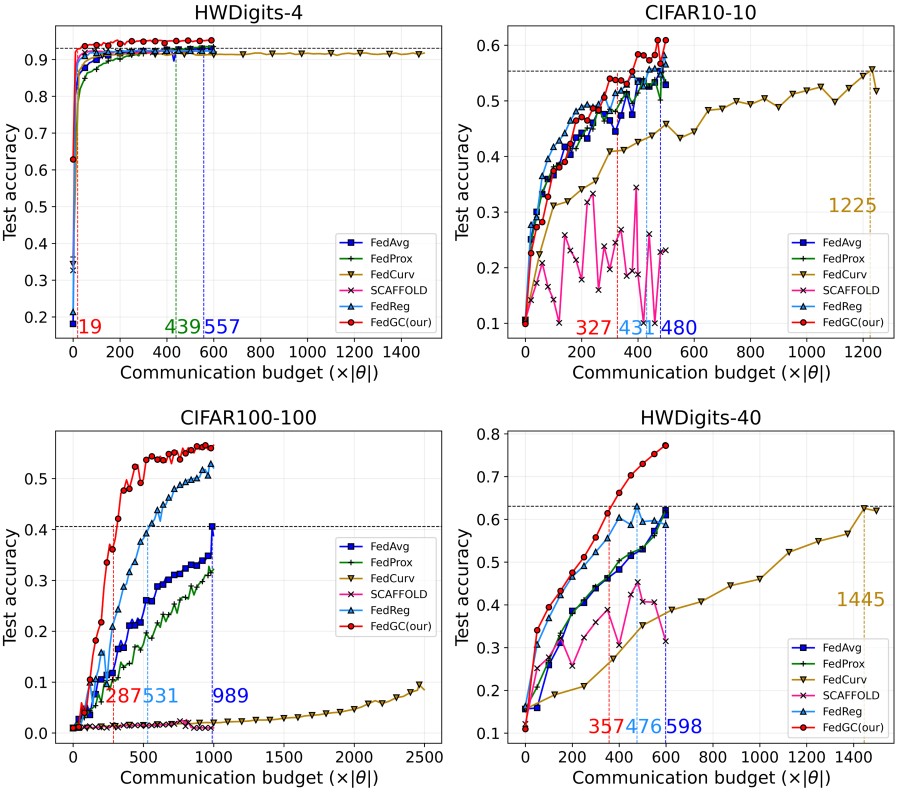

## A.9 ABLATION EXPERIMENTS

We conduct the ablation experiment to disentangle CGC & SGC on all four compared datasets.

The results in Table 4 illustrate that the performances of our FedGC will degrade without CGC, in particular for the label skew datasets. That is because the issue of catastrophic forgetting seriously

Table 4: Ablation experiments.

| Method | HWDigits-4 | CIFAR10-10 | CIFAR100-100 | HWDigits-40 |
|---|---|---|---|---|
| FedAvg | 93.07 | 55.34 | 40.59 | 62.11 |
| FedReg | 92.32 | 57.70 | 53.02 | 63.05 |
| FedGC (w/o CGC) | 93.38 | 54.09 | 41.64 | 74.51 |
| FedGC (w/o SGC) | 94.15 | 59.64 | 55.73 | 74.63 |
| FedGC | 95.37 | 60.92 | 56.56 | 77.33 |

influences the performance of FL, and our CGC can mitigate this forgetting issue. Therefore, in our FedGC, CGC is very important to improve the performance of FL on Non-IID data.

### A.10 AVERAGE LOSSES OF EACH CLIENT ON GLOBAL DATA BEFORE AND AFTER LOCAL TRAINING IN FEDAVG

Here we give an example to showcase that the forgetting issue is important. For FedAvg on CIFAR10-10, we calculate $L^{t-1}$ and $L_k^t$ to show the average losses of each client on global data before and after local training respectively.

$$L^{t-1} = \frac{1}{|K|} \sum_{k=1}^{K} L(\theta_k^{t-1}, \mathcal{D}) \tag{34}$$

where $K$ is the number of clients, $\theta_k^{t-1}$ is the initial model before local training, $\mathcal{D}$ is the global data.

$$L_k^t = \frac{1}{|K|} \sum_{k=1}^{K} \frac{1}{|K|} \sum_{i=1}^{K} L(\theta_i^t, \mathcal{D}_k) \tag{35}$$

where $\theta_i^t$ is the local model after local training and $\mathcal{D}_k$ is the local data.

Figure 7: Average losses of each client on global data before ($L^{t-1}$) and after ($L_k^t$) local training in FedAvg, and each point indicates one communication round.

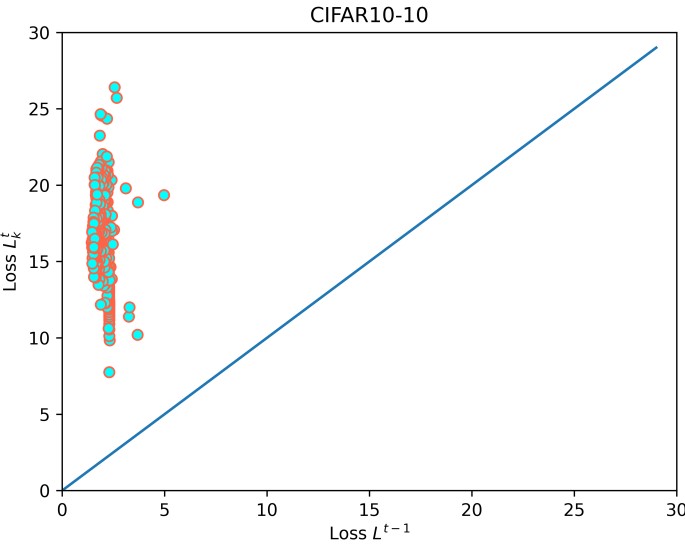

The Figure 7 shows that in FedAvg, the average losses $L_k^t$ on global data for all communication rounds are significantly larger than $L^{t-1}$ after local training. This is because the clients forget the knowledge learned from global model after local training in FedAvg.

