# OpenReview forum: "FedGC: An Accurate and Efficient Federated Learning under Gradient Constraint for Heterogeneous Data"
_ICLR.cc/2023/Conference — Submitted to ICLR 2023_

### Official Review · Reviewer_yV7y · 2022-10-23

**Confidence:** 4
**Clarity, Quality, Novelty And Reproducibility:** 1. I don't quite follow the issue of …
**Correctness:** 4
**Technical Novelty And Significance:** 2
**Empirical Novelty And Significance:** 3
**Recommendation:** 6

**Strength And Weaknesses:**

Strength:
1. The proposed methods are intuitive, easy to understand and implement with the aid of QP solver.
2. The proposed methods are very effective in experiments.

Weakness:
1. There are no explanations on the development of these methods, maybe some examples in image classification tasks can be given, or give some explanations to persuade people that these methods may also work in NLP tasks that are not experimented in this paper.
2. There is no theoretical support for the proposed methods.
3. The QP constrained optimization orientated gradient projection idea is not new in the literature (such as https://arxiv.org/pdf/2101.11296.pdf), some more comparisons may be needed in terms of novelty of the proposed methods.

**Summary Of The Paper:**

 This paper proposes a set of new methods to deal with the "catastrophic forgetting" issue for federated learning over heterogeneous data.  The main idea of this method is to project the newly sampled gradients into the set with acute angle with previous gradients which is kind of like efficient regularization that has an efficient QP solution. The boost in performance is significant in that both the reported test accuracy and communication cost required to reach certain accuracy demonstrate superiority in contrast to existing methods.

**Summary Of The Review:**

This paper crafts a set of methods that demonstrates great improvement in experiments to deal with the "catastrophic forgetting" issue in federated learning over heterogeneous datasets. Given that other methods are also well tuned, the test accuracy of the proposed methods have some edges towards existing methods and the reported communication cost and training time savings are significant.

---

> ### Author Response · Authors · 2022-11-17
> **Response to Reviewer yV7y (Part 1/2)**
>
> Dear Reviewer,
>
> We thank the reviewer for providing helpful feedback to improve our work. We address concerns and questions from the reviewer, and answer them individually.
>
> > **Q1: There are no explanations on the development of these methods, maybe some examples in image classification tasks can be given, or give some explanations to persuade people that these methods may also work in NLP tasks that are not experimented in this paper.**
>
> >**Q4: I don't quite follow the issue of "catastrophic forgetting", may be more illustrations are needed in the introduction, the current explanations to this issue seems to me too short and too simple, may be the authors can give an example to showcase that the forgetting issue is important and what aspects of this issue motivate the proposed methods.**
>
> **Response:** Since these two comments are all about the explanations on the development of our method, we respond them together as shown below.
>
> The explanations on the development of our method are shown below. If the data are Non-IID across clients, the local optimum of each client can be far from the others after local training and the initial model parameters received from server will be overridden. Hence, the clients will forget the initially received knowledge from the server, i.e., the clients suffer from the *catastrophic forgetting* of the learned knowledge from other clients. In other words, there is a drastic performance drop (or loss increase) of model on global data after local training. Here we give an example to showcase that the forgetting issue is important in **Appendix A.10**. For FedAvg on CIFAR10-10, we calculate $L^{t-1}$ and $L_k^t$ to show the average losses of each client on global data before and after local training respectively.
>
> $$L^{t-1} = \frac{1}{|K|} \sum_{k=1}^K L(\theta_k^{t-1}, \mathcal{D})$$
>
> where $K$ is the number of clients, $\theta_k^{t-1}$ is the initial model before local training, $\mathcal{D}$ is the global data.
> $$L_k^t = \frac{1}{|K|} \sum_{k=1}^K \frac{1}{|K|} \sum_{i=1}^K L (\theta_i^t, \mathcal{D}_k)$$
> where $\theta_i^t$ is the local model after local training and $\mathcal{D}_k$ is the local data.
>
>
> The **Figure 7** in **Appendix A.10** shows that in FedAvg, the average losses $L_k^t$ on global data for all communication rounds are significantly larger than $L^{t-1}$ after local training. This is because the clients forget the knowledge learned from global model after local training in FedAvg. This motivates this study for a new approach of mitigating catastrophic forgetting. And the theorical verification in **Appendix A.3** has shown that through our CGC, the loss of model on global data will reduce after local training.
>
>
> Actually, our FedGC is quite general and can also be applied to NLP tasks by replacing the computer vision model with NLP models (e.g., LSTM)
>
>
> > **Q2: There is no theoretical support for the proposed methods.**
>
>
> **Response:** Thanks for the comment, we have provided the theoretical analysis of gradient projection in **Appendix A.3** and convergence analysis in **Appendix A.4**.

---

> > ### Author Response · Authors · 2022-11-17
> > **Response to Reviewer yV7y (Part 2/2)**
> >
> > > **Q3: The QP constrained optimization orientated gradient projection idea is not new in the literature (such as https://arxiv.org/pdf/2101.11296.pdf), some more comparisons may be needed in terms of novelty of the proposed methods.**
> >
> >
> > **Response:** Thanks for the suggestion, in the literature [1] (suggested by the reviewer), FedH2L is proposed. To perform gradient projection, ***the shared public seed data*** are necessary in FedH2L. In this way, it can calculate $g_i^{pub}$ and then performs gradient projection (see Eq. (8) in [1]). Thus, although FedH2L uses gradient projection, it requires one public dataset shared between clients, ***which seriously undermines the privacy-preserving***.
> >
> > Actually, **FedReg** [2] can also be considered as gradient projection method, as shown in Eq. (8) and (9) in their paper. However, it needs to generate pseudo data at clients, which increases a lot of computational and storage costs for clients.
> >
> > $$ \theta^{(t,i)'} = argmin_{\theta} ||\theta - \theta^{(t, i)}||^2  s.t. (\theta^{(t-1)} - \theta)^T g_{\beta}(\mathbb{D}_i^s) \geq 0 (8)$$
> >
> > $$ \theta^{(t,i)*} = argmin_{\theta} ||\theta - \theta^{(t, i)'}||^2  s.t. (\theta^{(t-1)} - \theta)^T g_{\beta}(\mathbb{D}_i^p) \geq 0 (9)$$
> >
> >
> >
> > Different from FedH2L and FedReg, our CGC in FedGC only utilizes the server gradient (i.e., the aggregated gradient from all clients) to perform gradient projection, and ***it is unnecessary to share data between clients or generate pseudo data***. Moreover, our FedGC can ensure privacy-preserving, even when the recently proposed gradient inversion attack DLG performs, as detailed in Appendix A.5 of the revised manuscript. Furthermore, besides CGC, we also proposed PGD and SGC to mitigate the Non-IID problem in FL.
> >
> > Since FedH2L focuses on the applications that participants may require heterogeneous network architectures and FedReg is more related to our work, only FedReg is used for comparison. In the experiments, compared to FedReg, our FedGC achieves higher accuracy but takes less training time and communication rounds.
> >
> >
> >
> > [1] Li Y, Zhou W, Wang H, et al. FedH2L: Federated learning with model and statistical heterogeneity[J]. arXiv preprint arXiv:2101.11296, 2021.
> >
> > [2] Chencheng Xu, Zhiwei Hong, Minlie Huang, and Tao Jiang. Acceleration of federated learning with alleviated forgetting in local training. In International Conference on Learning Representations, 2022.
> >
> >
> > > **Q5: No code is provided for reproducing the results.**
> >
> >
> > **Response:** After we make the code clean and readable and add some testing cases, we will upload our code.
> >
> >
> > > **Q6: In section "Step 1", it seems to me Eg―kt=g~kt, so they should be on the same line in Figure 1, can you explain this a little bit?**
> >
> >
> > **Response:** We apologize for mis-using symbols, and correct this error in Step 1 of page 4 by replacing the statement
> > "Instead of the mean gradient $\bar{g}_k^t$=
> >
> > $\frac{1}{E}\sum_{l=1}^{E} g^t_{k,l}$ used in Minibatch SGD"
> > with
> > "Different from the mean gradient $\bar{g}_k^t$=
> >
> > $\frac{1}{B}\sum_{b=1}^{B}g^t_{k,b'}$ used in Minibatch SGD that averages multiple gradients for several mini-batches data at the same point (e.g., $\theta_{i,0}^t$)", in the revised manuscript. In addition, **Figure 1** has been revised in the the revised manuscript by adding the calculation of mean gradient $\bar{g}_k^t$ (in blue dot rectangle boxes) in Minibatch SGD, to distinguish it from our pseudo gradient $\tilde{g}_k^t$.

---

> > > ### Comment · Reviewer_yV7y · 2022-11-29
> > > **Reply to the rebuttal**
> > >
> > > I thank the authors for the replies. I went over the theoretical analysis newly added, but I feel the assumption there is way stronger than it should be. On the other hand, I still think this work has its algorithmic contributions. I have no more questions and comments.

---

> > > > ### Author Response · Authors · 2022-11-30
> > > > **Response to: Reply to the rebuttal**
> > > >
> > > > We thank Reviewer yV7y again for providing a valuable pre-rebuttal review. Your detailed suggestions have greatly helped us in the paper revision.

---

### Official Review · Reviewer_NKr4 · 2022-10-24

**Confidence:** 4
**Correctness:** 3
**Technical Novelty And Significance:** 3
**Empirical Novelty And Significance:** 3
**Recommendation:** 3

**Clarity, Quality, Novelty And Reproducibility:**

Clarity is high, quality needs some improvement, the proposed methods are original.
Reproducibility seems alright, I am confident I could replicate these results given the presented hyperparameters, algorithms and explanation. I would encourage the authors to provide python-code in the appendix detailing the application of quadprog within their projection schemes.

**Strength And Weaknesses:**

The paper's strength lies in the contribution through CGC and SGC. These modified update schemes are well motivated and show good empirical results.
The paper's weakness lies in some of the algorithmic details & discussion:
The discussion around 'Step 1 (PDG)' seems ad-hoc and not well-founded. Traditionally, FedAvg doesn't consider Mini-batch SGD since generally speaking FL tries to capitalize on the multiple local update steps in order to save on communication and decrease overall algorithm run-time. As such, this point doesn't seem worthwhile to explicitly make. The definition of $\bar{g}_k^t$ for the 'mean gradient' of Mini-batch SGD seems to suggest an average of gradients, each of which is computed on a different models through intermediate application of the individual gradient steps. My understanding of Mini-batch SGD is that a client's gradient is computed on the same model and the final update corresponds to the gradient of a very large mini-batch. The update that the authors propose for $\tilde{g}_k^t$ seems to be the standard (adaptive) FedAvg update that a client sends to the server (here scaled by $\eta$). See e.g. Algorithm 1 in 'Reddi, Sashank, et al. "Adaptive federated optimization." arXiv preprint arXiv:2003.00295 (2020).'

I would need some clarification on the details in Algorithm 2 `ClientLocalUpdate`:
The server-side code suggests that you explicitly consider client sub-sampling (which is great!). As a consequence, you need to explicitly send the previous server-side model $\theta_k^{t-1}$ along with the gradient $g^{t-1}$ to reconstruct the current server-side model $\theta_{k,0}^t=\theta_{k}^t$. In principle, this prevents you from using server-side adaptive optimizers that rely on momenta (such as FedAdam in above Reddi et al. paper). Ideally, you communicate $\theta_{k}^t$ directly to the client. Your algorithm explicitly states that you are communicating both, $g^{t-1}$ and  $\theta_k^{t-1}$  to the client, however you mention "Moreover, our FedGC only transmits the gradients between the server and clients." (also in the discussion: " Moreover, FedGC enables low communication cost due to fast convergence rate and only transmissions of gradients between server and clients"). Furthermore, you critique SCAFFOLD for doubling communication cost from server to client). In Section 4.4 you deduce that "the average gradient in the previous communication round is less useful in current round". This is curious to me, since your method also communicates the previous (projected) gradient - and uses it in a different way than Scaffold. Can you make a more precise statement here on why you believe your method is less susceptible to the use of previous-round gradient?
Further in section 4.4 you are critiquing "SCAFFOLD also requires more time to calculate and transmit the control variate between clients and server." Again, your method requires computing and sending the projected gradient to clients, in addition to model parameters.
Based on your somewhat contradicting discussion and lack of explicit statement about the communication-costs of FedGC, I have low confidence in your experimental evaluation as regards the communication costs (Table 2 & Section 4.5). I believe your method communicates $2.0 |\theta\$ from server to client (same as scaffold). Can you confirm that? And did you account for that in Table 2 correctly? I would like to see a learning curve similar to Figure 3, where the x-axis is scaled by communication-budget (i.e in multiples of $|\theta\$) instead of communication rounds to substantiate your claims.

Why did you choose to apply gradient projection layer-wise and not align the whole concatenated gradient? I am not sure I understand the argument that different layers have different dimension? How would that influence the flatten, then concatenate alternative?

Experiments:
related to above discussion: Are you using client-subsampling? How many clients are selected per-round?
Why are you using $E=50$ for FedGC and not the base-line? Please show an ablation study for these different settings. If using $E=50$ is required for FedGC, the computational cost is increased many times and claiming FedGC is well-suited because of low computation costs (as in the discussion) is misleading.

Another ablation I think would be very insightful to disentangle CGC & SGC further is to run without CGC and only have server-side projection. Such a method would come 'for free' in the sense that there is no communication overhead server->client.


Nit-picking:
Your last sentence in the main-body states "In a nutshell, FedGC has great potential for many real-world applications that concerns performance, real-time, communication & computational costs, and privacy preservation." I would like to see claims about 'real-time' made more concrete and substantiated, as well as how FedGC is privacy-preserving, since e.g. is it amenable to classical Differential privacy techniques (I believe it should be) and arguably reveals to clients the average update of previous-round clients, which is more privacy-revealing than standard FedAvg.

**Summary Of The Paper:**

The authors propose modifications to the FedAvg algorithm: CGC & SGC. The CGC gradient projection empirically improves convergence by aligning each client's update with the server-side update. SGC attempts to improve upon the weighted clients' average by projecting it to the individual clients' updates. The paper has some open remaining questions in the empirical section and some corrections required for 'PDG' imo.


**Summary Of The Review:**

Interesting algorithm with some weakness in the discussion (c.f. PDG), details (communication, subsampling) and the experimental evaluation.
Should the authors improve their discussion around PDG, and provide the missing information about communication budget and client subsampling, I will raise my score. Should the authors also provide additional experiments, I will raise my score again.


---- post-rebuttal ----
I have read the rebuttal and other reviewer's comments and as a consequence have reduced my score to 3

---

> ### Author Response · Authors · 2022-11-17
> **Response to Reviewer NKr4 (Part 1/4)**
>
> Dear Reviewer,
>
> We thank the reviewer for providing helpful feedback to improve our work. We address concerns and questions from the reviewer, and answer them individually.
>
> > **Q1.1: The discussion around 'Step 1 (PDG)' seems ad-hoc and not well-founded. Traditionally, FedAvg doesn't consider Mini-batch SGD since generally speaking FL tries to capitalize on the multiple local update steps in order to save on communication and decrease overall algorithm run-time. As such, this point doesn't seem worthwhile to explicitly make.**
>
> **Response:** We agree with the reviewer that FedAvg doesn’t adopt mini-batch SGD due to saving communication costs. However, when data are Non-IID, too many iterations in local training may cause local models to be biased towards their local data and enlarge the difference between the local model and the global model. This is not conducive to the convergence of global model. In the revised manuscript, we strengthen the motivation of the proposed PGD. Moreover, we conduct the experiment of FedAvg adopting mini-batch SGD on HWDigits-4 (Attribute skew) dataset. The number of mini-batches are in range {1, 10, 20, 50}, and the number of communication rounds $T$ is 600 for all experiments. The experimental result is below
>
> |T=600|B=1|B=10|B=20|B=50| E=1(i.e., B=73~732)|
> |----|----|----|----|----|----|
> |HWDigits-4|85.22|92.75|93.43|93.59|93.07|
>
> We can see that FedAvg can achieve better performance when $B=50$ while requiring less iterations in local training stage compared to $E=1$. The experimental results are consistent with our statement.
>
> > **Q1.2: The definition of g¯kt for the 'mean gradient' of Mini-batch SGD seems to suggest an average of gradients, each of which is computed on a different models through intermediate application of the individual gradient steps. My understanding of Mini-batch SGD is that a client's gradient is computed on the same model and the final update corresponds to the gradient of a very large mini-batch.**
>
> **Response:** We apologize for mis-using symbols, and correct this error in Step 1 of page 4 by replacing the statement
> "Instead of the mean gradient $\bar{g}_k^t$=
>
> $\frac{1}{E}\sum_{l=1}^{E} g^t_{k,l}$ used in Minibatch SGD"
> with
> "Different from the mean gradient $\bar{g}_k^t$=
>
> $\frac{1}{B}\sum_{b=1}^{B}g^t_{k,b'}$ used in Minibatch SGD that averages multiple gradients for several mini-batches data at the same point (e.g., $\theta_{i,0}^t$)", in the revised manuscript. In addition, **Figure 1** has been revised in the the revised manuscript by adding the calculation of mean gradient $\bar{g}_k^t$ (in blue dot rectangle boxes) in Minibatch SGD, to distinguish it from our pseudo gradient $\tilde{g}_k^t$.
>
> > **Q1.3: The update that the authors propose for g~kt seems to be the standard (adaptive) FedAvg update that a client sends to the server (here scaled by η). See e.g. Algorithm 1 in 'Reddi, Sashank, et al. "Adaptive federated optimization." arXiv preprint arXiv:2003.00295 (2020).'**
>
> **Response:** In terms of formulas, we agree with the reviewer that the proposed $\tilde{g}_k^t$ is the one that a client sends to the server but scaled by $1/\eta$. The scaling in our FedGC can promote a large update for local model to accelerate the convergence rate. And the difference is that in FedAvg, each client adopts multiple epochs for local training, but in our FedGC, each client is trained by multiple mini-batches. Hence, the step 1 (PGD) includes two parts: mini-batch update and pseudo-gradient. For clarity, we have revised the step 1 in the revised manuscript to make it well-founded.

---

> > ### Author Response · Authors · 2022-11-17
> > **Response to Reviewer NKr4 (Part 2/4)**
> >
> > > **Q2.1:  I would need some clarification on the details in Algorithm 2 ClientLocalUpdate: The server-side code suggests that you explicitly consider client sub-sampling (which is great!). As a consequence, you need to explicitly send the previous server-side model θkt−1 along with the gradient gt−1 to reconstruct the current server-side model θk,0t=θkt. In principle, this prevents you from using server-side adaptive optimizers that rely on momenta (such as FedAdam in above Reddi et al. paper).**
> >
> >
> > **Response:** Sorry for confusing the reviewer. Actually, ONLY the gradient $g^{t-1}$ is necessary to be transferred between clients and server, and we have revised the pseudo code **Algorithm 1** of our FedGC to avoid confusion.
> >
> > Our FedGC can randomly sample a subset of clients to participate in local training (Lines 9-18 in **Algorithm 1**) and aggregate their gradients $g_k^t, k \in \mathcal{K}$ on server (Lines 19-21). Then server broadcasts the aggregated $g^t$ to all clients to synchronize their parameters (Lines 3-8). For each client, only $g^{t-1}$ is received from server (Line 6) and it sends $g_k^t$ to server (Line 16). $\theta^t_{k}$ is maintained locally and does not require to transfer between the server and clients. Because of only transferring gradient $g^{t-1}$ from server to clients, we can utilize other optimizers (including optimizers with momenta, e.g., FedAdam) to update parameters at server (Line 20) and client (Line 7) beside SGD, only ensuring the optimizers are same.
> >
> > > **Q2.2: Ideally, you communicate θkt directly to the client. Your algorithm explicitly states that you are communicating both, gt−1 and θkt−1 to the client, however you mention "Moreover, our FedGC only transmits the gradients between the server and clients." (also in the discussion: " Moreover, FedGC enables low communication cost due to fast convergence rate and only transmissions of gradients between server and clients"). Furthermore, you critique SCAFFOLD for doubling communication cost from server to client).**
> >
> > **Response:** Referring to the response of **Q2.1** and **Algorithm 1**, our FedGC only transfers the gradient $g^{t-1}$ between server and clients. Therefore, the communication costs of our FedGC is $|\theta|$ in each communication round, where $|\theta|$ represents the model parameters and it has the same size with gradients.
> >
> >
> > > **Q2.3: In Section 4.4 you deduce that "the average gradient in the previous communication round is less useful in current round". This is curious to me, since your method also communicates the previous (projected) gradient - and uses it in a different way than Scaffold. Can you make a more precise statement here on why you believe your method is less susceptible to the use of previous-round gradient?**
> >
> >
> > **Response:** Sorry for confusing the reviewer, we have corrected this statement for clarity in Section 2 and 4.3.
> > > “the average gradient in the previous communication round may not satisfied its assumption that $c_j$ $\approx g_j(y_i)$ and $c \approx \frac{1}{N} \sum_{j}g_j(y_i)$, especially when deep learning models on image datasets perform at clients [1]”
> >
> > The detailed explanations are shown below. Scaffold wants to obtain the ideal update on client $i$
> >
> > $y_i \leftarrow y_i + \frac{1}{N}\sum_jg_j(y_i)$ &#160;(6)
> >
> > by correcting the "client-drift" with server control variate $c$ and client control variate $c_i$ as shown below
> >
> > $y_i \leftarrow y_i - \eta_l(g_i(y_i) - c_i + c)$ &#160;(3)
> >
> > Next, we briefly introduce the training procedure of Scaffold.
> >
> > At the *0-th* round, $y_i = x_0$, $c_i=c=0$, where $y_i$ is the local model of client $i$, $x_0$ is the server model.
> >
> > &#160;\# at client $i$
> >
> > &#160;&#160;for $k= {1, \dots K}$
> >
> > &#160;&#160;&#160; $y_i \leftarrow y_i - \eta_l(g_i(y_i))$
> >
> > &#160;&#160; $c_i = g_i(x_0)$
> >
> > &#160; \# at server
> >
> > &#160;&#160; $c = \frac{1}{N}\sum_ic_i$
> >
> > &#160;&#160; $x_1 \leftarrow Server Aggregation$
> >
> > At the *1-st* round, client $i$ receives $x_1$ from server and then perform local training:
> >
> > &#160; When $k = 1$:
> >
> > &#160;&#160; $y_i=x_1-\eta_l(g_i(x_1) - g_i(x_0) + \frac{1}{N}\sum_jg_j(x_0))$
> >
> > &#160; When $k = 2:K$:
> >
> > &#160;&#160; $y_i=y_i-\eta_l(g_i(y_i) - g_i(x_0) + \frac{1}{N}\sum_jg_j(x_0))$
> >
> > Obviously, the assumptions that $c_j$ $\approx g_j(y_i)$ and $c \approx \frac{1}{N}\sum_{j}g_j(y_i)$ may not be satisfied, especially for deep models on image datasets, as verified by literature [1].
> >
> >
> > While in our CGC, the previous-round gradient $g^{t-1}$ is used to ensure that the loss at global data reduces after local update. The theorical verification is detailed in Appendix A.3. Hence, even for deep learning models on image datasets, the gradient projection based on previous-round global gradient can work well.
> >
> > [1] Li Q, He B, Song D. Model-contrastive federated learning[C]//Proceedings of the IEEE/CVF Conference on Computer Vision and Pattern Recognition. 2021: 10713-10722.

---

> > > ### Author Response · Authors · 2022-11-17
> > > **Response to Reviewer NKr4 (Part 3/4)**
> > >
> > > > **Q2.4: Further in section 4.4 you are critiquing "SCAFFOLD also requires more time to calculate and transmit the control variate between clients and server." Again, your method requires computing and sending the projected gradient to clients, in addition to model parameters. Based on your somewhat contradicting discussion and lack of explicit statement about the communication-costs of FedGC, I have low confidence in your experimental evaluation as regards the communication costs (Table 2 & Section 4.5). I believe your method communicates 2.0|θ from server to client (same as scaffold). Can you confirm that? And did you account for that in Table 2 correctly? I would like to see a learning curve similar to Figure 3, where the x-axis is scaled by communication-budget (i.e in multiples of |θ) instead of communication rounds to substantiate your claims.**
> > >
> > > **Response:** Referring to the responses of **Q2.1**, **Q2.2** and **Algorithm 1**, our FedGC only transfer gradients  between server and clients. Therefore, the communication cost of FedGC is $|\theta|$ at each round, not $2 \times |\theta|$. The results in Table 2 are correct. Following the suggestion of the reviewer, the learning curves are added in **Appendix A.8**, where the x-axis is scaled by communication-budget. The figure shows that our FedGC can reach the target accuracy (i.e., the highest accuracy of baseline FedAvg) with the smallest communication budget.
> > >
> > > > **Q3: Why did you choose to apply gradient projection layer-wise and not align the whole concatenated gradient? I am not sure I understand the argument that different layers have different dimension? How would that influence the flatten, then concatenate alternative?**
> > >
> > > **Response:** Sorry for confusing the reviewer, we have revised the explanation why layer-wise manner is performed for gradient projection, as shown below.
> > >
> > > "Since the whole concatenated gradient has very large dimension, we iteratively perform the gradient projection layer by layer (i.e., layer-wise manner) to reduce the memory overhead. "
> > >
> > > In the following, we will use an example to explain it in detail. For example, ResNet-9 used in our experiments contains $3\times3\times3\times64=1728$ parameters (as the same as gradients) in its first layer and $3\times3\times64\times128=73728$ in its second layer, and so on. The way of concatenating all flattening gradients in ResNet-9 into a gradient vector will cause the dimension of variables in Eqs.(6) and (9) being huge ($p = 1728+ 73728+...$) and dramatically increases the memory space for gradient projection. In contrast, the layer-wise  manner in our work is very efficient and does not require a large memory space.
> > >
> > >
> > > > **Q4.1: Experiments: related to above discussion: Are you using client-subsampling? How many clients are selected per-round?**
> > >
> > > **Response:** In our experiment, all clients participate in the local training and aggregation, and the client fraction is 1. Since the HWDigits-4 and CIFAR10-10 only contain 4 and 10 clients respectively, it is unnecessary to set the fraction less than 1. Hence, for consistency, we set client fraction equals 1.0 in all four datasets. But our FedGC supports client-subsampling as shown in **Algorithm 1** (Lines 9-18). We add the ablation experiment about client fraction on HWDigits-4 with FedAvg and our FedGC. The experimental results are shown below.
> > >
> > >
> > > | Client fraction|0.5| 1.0|
> > > |-|-|-|
> > > |FedAvg|71.88|93.07|
> > > |FedGC(our)|91.41|**95.27**|
> > >
> > > When fraction equals to 0.5, that means only two clients participate in local training and aggregation in each round, and the performance will be decreased. The results in the above table show that there is a drastic performance drop on FedAvg with fraction=0.5, while our FedGC still achieves satisfactory accuracy. That is mainly benefited from our proposed CGC and SGC.

---

> > > > ### Author Response · Authors · 2022-11-17
> > > > **Response to Reviewer NKr4 (Part 4/4)**
> > > >
> > > > > **Q4.2: Why are you using E=50 for FedGC and not the base-line? Please show an ablation study for these different settings. If using E=50 is required for FedGC, the computational cost is increased many times and claiming FedGC is well-suited because of low computation costs (as in the discussion) is misleading.**
> > > >
> > > >
> > > > **Response:** We apologize for confusing the reviewer that we use a symbol $E$ to represent different definitions in FedGC (our) and other compared methods in the original manuscript.
> > > >
> > > > Our FedGC adopts ***multiple mini-batches*** to train local models at each communication round, while other compared methods train their local models by performing ***multiple epochs*** on full local data. For clarity, we define $B$ as the number of mini-batches used in our FedGC and define $E$ as the number of epochs in other compared methods. Hence, in the experiments of our FedGC on HWDigits, $B=50$, not $E=50$.
> > > >
> > > > In addition, we conduct the ablation experiments of our FedGC on HWDigits-4 under different $B$. The experimental results are shown below. The results show that under $B=50$, our FedGC can achieve satisfactory accuracy with less local training time.
> > > >
> > > > |B|20|50|100|
> > > > |-|-|-|-|
> > > > |Acc.|93.34|**95.27**|93.84|
> > > > |Time(s)|2.17|**6.33**|11.69|
> > > >
> > > >
> > > > > **Q5: Another ablation I think would be very insightful to disentangle CGC & SGC further is to run without CGC and only have server-side projection. Such a method would come 'for free' in the sense that there is no communication overhead server->client**
> > > >
> > > >
> > > > **Response:** Thanks for the reviewer's suggestion. We conduct the ablation experiment to disentangle CGC & SGC on four datasets. The experimental results are shown below.
> > > >
> > > > | | HWDigits-4 | CIFAR10-10 | CIFAR100-100 | HWDigits-40 |
> > > > |-|-|-|-|-|
> > > > | FedAvg| 93.07 | 55.34 | 40.59 | 62.11  |
> > > > | FedReg| 92.32  | 57.70 | 53.02  | 63.05 |
> > > > | FedGC (w/o CGC) | 93.38 | 54.09 | 41.64 | 74.51 |
> > > > | FedGC (w/o SGC) | 94.15| 59.64 | 55.73 | 74.63 |
> > > > | FedGC | 95.37 | 60.92 | 56.56 | 77.33 |
> > > >
> > > > The results illustrate that the performances of our FedGC will degrade without CGC, in particular the label skew datasets. That is because the issue of catastrophic forgetting seriously influences the performance of FL, and our CGC can mitigate this forgetting issue. Therefore, in our FedGC, CGC is very important to improve the performance of FL on Non-IID data.
> > > >
> > > >
> > > >
> > > > > **Q6: Nit-picking: Your last sentence in the main-body states "In a nutshell, FedGC has great potential for many real-world applications that concerns performance, real-time, communication & computational costs, and privacy preservation." I would like to see claims about 'real-time' made more concrete and substantiated, as well as how FedGC is privacy-preserving, since e.g. is it amenable to classical Differential privacy techniques (I believe it should be) and arguably reveals to clients the average update of previous-round clients, which is more privacy-revealing than standard FedAvg.**
> > > >
> > > > **Response:** Thanks for the comment. Since our FedGC has small training time and fast convergence rate with less communication costs, it is friendly to some real-time applications.
> > > >
> > > > We have provided the privacy-preserving analysis of our FedGC using one popular gradient inversion attack “Deep Leakage from Gradients (DLG)” [1] in Section **Appendix A.5**, and conduct the gradient inversion attacks experiment to compare FedAvg and our FedGC with DLG on CIFAR10-10 dataset. The analysis and experimental result show that our FedGC can ensure the preserving of privacy.
> > > >
> > > > In addition, as stated in FedReg, “Differential privacy (DP) is one of the most widely used strategies to prevent the leakage of private information. However, when DP is incorporated into FL, the performance of the resulting model decays significantly.” Hence, DP is not incorporated into our FedGC, and our FedGC without DP already can ensure preserving of privacy.
> > > >
> > > >
> > > > [1] Ligeng Zhu, Zhijian Liu, and Song Han. Deep leakage from gradients. In H. Wallach, H. Larochelle, A. Beygelzimer, F. d'Alch´e-Buc, E. Fox, and R. Garnett (eds.), Advances in Neural Information Processing Systems, volume 32. Curran Associates, Inc., 2019.
> > > >
> > > >
> > > > > **Q7: I would encourage the authors to provide python-code in the appendix detailing the application of quadprog within their projection schemes.**
> > > >
> > > >
> > > > **Response:** Thanks for the reviewer's good suggestion. We provide the reviewer the python code snippet of QP as follow
> > > >
> > > > ```
> > > > # grad_tilde: input gradient vector
> > > > # G: the matrix G in Eq. (9) for SGC, and $g^{t-1}$ for CGC in Eq. (5)
> > > > # C: small positive constant C in Eqs.(5) and (9)
> > > >
> > > > t = G.shape[0]
> > > > P = np.matmul(G, G.transpose())
> > > > P = 0.5 * (P + P.transpose()) + np.eye(t) * 1e-3
> > > > q = (np.matmul(G, grad_tilde) - C) * -1
> > > > G1 = np.eye(t)
> > > > h = np.zeros(t)
> > > > v = quadprog.solve_qp(P, q, G1, h)[0]
> > > > GTv = np.matmul(G.transpose(), v)
> > > >
> > > > grad = grad_tilde + GTv
> > > >
> > > > ```

---

> ### Comment · Reviewer_NKr4 · 2022-11-18
> **Thank you for the rebuttal**
>
> I have read your rebuttal, as well as the other reviews.
> I appreciate the effort that went into this extensive rebuttal, however some points clearly remain:
> - Presentation of 'Step 1'
> - The communication budget
>
> These have not been addressed sufficiently and, among some other points also raised by reviewer ESza should be fixed before this paper is ready for publication.
>
> In Step 1, you discuss the 'Mini-batch update'. There is quite some discussion around how many local update steps your method and other methods perform. This is out-of-place for this paper for two reasons. Firstly, the number of local update steps is a hyper-parameter that has to be considered in any FL application, both in-simulation as well as in real applications. It is a largely arbitrary choice whether one performs updates in multitudes of $E$ (Epochs) or mini-batches $B$. In the end, the hyperparameter determines a potentially different number of steps per client depending on the data-set size. Depending on the degree of non-iid-ness of the data-set and how different data-sets are in size, the performance is impacted one way or the other by the choice of number of update steps. This is an observation that is orthogonal to your paper and out-of-place. It doesn't contribute any novel insights, beyond an interesting empirical case-study in Appendix A.7 for a specific data-set.
>
> Crucially, the choice of number of local updates is orthogonal to any federated algorithm you consider and should be treated as a hyper-parameter for all of them, either fixed across all baselines or separately fine-tuned for an improved empirical comparison.
> At the moment, this discussion is just providing confusion and gives the impression of a claim of novelty that isn't there.
>
> Furthermore, the discussion around the Pseudo-gradient is equally mis-leading. Equation (4) is exactly equivalent to what FedAvg (and related algorithms) is all about. Scaling the pseudo-gradient by the learning rate is equivalent to choosing a different learning-rate for the server-side update in generalized FedAvg (Reddit et al., 2020).
>
> You have modified Algorithm 1, which clearly uncovers your communication strategy as far as I can see. In every single communication round, you are broadcasting $g^{t-1}$ to every single client in the federation. Crucially, not every client in the per-round participating set of clients, but every single client in the federation. Your communication budget therefore scales with $|K|$ as opposed to $\mathcal{K}$, making this algorithm entirely infeasible for cross-device FL with large federations.
> Further in practice such an algorithm will be infeasible because it requires every client to be available all the time. An equivalent approach to Algorithm 1 is to only communicate with the clients in $\mathcal{K}$ and send them both, $g^{t-1}$ and $\theta^{t-1}$, thereby doubling the communication from server to client. This strategy is equivalent to what Scaffold is doing for cross-silo FL applications.
>
> Figure 6 is mis-leading, as apparently you're not using client-subsampling in your experiments.
> Please show experiments for sub-sampling and provide your communication-budget analysis for e.g. the stackoverflow-logistic-regression task presented in (reddi et al., (2020)), which has about 340.000 clients with 50 clients/round.
>
> Let me make a final statement about privacy in FL. While it is commendable that you show results on a gradient inversion attack, this does not "*ensure preserving of privacy*". The only current **guarantee** of privacy is given through Differential Privacy and your text needs to reflect this.
>
> I thank the author for clarifying their algorithm and providing this discussion. Unfortunately given these insights and mismatch of claims I need to reduce my score and cannot suggest this paper for publication in its current form.

---

> > ### Author Response · Authors · 2022-11-19
> > **Response to: Thank you for the rebuttal**
> >
> > Thank you for taking the time to consider our responses. First of all, we would like to emphasize that the research objective of this work is to propose ***an accurate and efficient (or fast convergent) Federated Learning under Gradient Constraint for Heterogeneous Data***, as given by the title.
> >
> > From the viewpoints of improved accuracy and efficiency of FL for heterogenous data, our work is of significant novelty and contributions as confirmed by the reviewer’s statement that ***“The paper's strength lies in the contribution through CGC and SGC. These modified update schemes are well motivated and show good empirical results”*** in the previous feedback.
> >
> > Nevertheless, the additional questions raised by the reviewer are about hyperparameters, privacy-preserving, and subset of clients. These challenging issues commonly exist and mostly not yet resolved in all FL algorithms including our newly proposed FedGC. Moreover, they are out of scope of our work and our method is not originally designed to address them. In previous rebuttal, we tried our best to provide additional explanations and experiments to answer the reviewer questions. Although the answers / results cannot completely fulfill / address the reviewer concerns, our rebuttal shows that our method is potential to address these issues in the future. Humbly speaking, these challenging issues require intensive effort to invent specific mechanisms to effectively address them. It is nontrivial and even conflicting to design a single mechanism to address ALL issues (accurate, efficient, no hyperparameter, privacy-preserving, low budget, subset of clients) in one time. Therefore, based on the reviewer’s insightful comments, we will continue to work hard for these issues in our future research. Hope the reviewer can kindly reconsider the score of the review again.
> >
> > In addition, in the final submission, we will emphasize the following points in order to avoid confusion and accurately reflect the limitations of our work:
> > - About scaling: We will emphasize that “Scaling the pseudo-gradient by the learning rate is equivalent to choosing a different learning-rate for the server-side update in generalized FedAvg (Reddit et al., 2020)” in Step1 of 3.1.
> > - About communication-budget: **When all clients participate**, our FedGC only transmits $g^{t−1}$ between server and clients, and Fig. 6 shows that our FedGC can reach the target accuracy (e.g., the highest accuracy of baseline FedAvg) with the smallest communication budget. **Limitation: When only subset of clients participates**, our FedGC requires every client to be available all the time, and if some clients are unavailable, both $g^{t−1}$ and $θ^{t−1}$ need to be transmitted. The computation budget will be **1.5(=3/2) $\times |\theta|$**, i.e., the server sends $g^{t−1}$ and $θ^{t−1}$ to the client $k$ and then client $k$ only send $g^{t}_k$ back to server after local training. **Notably**, although our FedGC has the above-mentioned limitation about communication-budget, it can achieve significantly higher accuracy when small subset of clients participates. For instance, when client fraction is 0.5, our FedGC gets 91.41% while FedAvg only gets 71.88% of accuracy for dataset HWDigits-4, as shown in following table.
> >
> > | |Client fraction = 0.5| Comm. cost|
> > |-|-|-|
> > |FedAvg|71.88| 277$\times$ \|$\theta$\| |
> > |FedGC(our)|91.41| 35 $\times$ 1.5 $\times$ \|$\theta$\||
> >
> > - About privacy-preserving: We agree that privacy-preserving is a large topic and there are numerous attacks that require specific privacy-preserving methods. Therefore, we removed the inappropriate claim “ensure preserving of privacy” and emphasize that “The analyses and experiments in A.5 show that our FedGC is more robust against gradient inversion attacks”. Moreover, we will emphasize that “The only current guarantee of privacy is given through Differential Privacy”.

---

> > ### Author Response · Authors · 2022-12-11
> > **Looking forward to your feedback.**
> >
> > Dear Reviewer NKr4,
> >
> > Thanks again for your valuable comments! We have responded to your recent comments in the section of "Response to: Thank you for the rebuttal". We are looking forward to your feedback and would be happy to address any further concerns you may have.
> >
> > Thank you, author

---

### Official Review · Reviewer_ESza · 2022-10-25

**Confidence:** 4
**Correctness:** 3
**Technical Novelty And Significance:** 2
**Empirical Novelty And Significance:** 2
**Recommendation:** 3

**Clarity, Quality, Novelty And Reproducibility:**

The paper is well written and easy to follow. The algorithm and the solution is original, but lacks any sort of theoretical guarantees and the empirical comparisons are not fair.

**Strength And Weaknesses:**

Strengths:
+ The paper is well written and easy to follow. The problem being considered is relevant and of sufficient interest to the FL community in general.
+ The method proposed utilizes inequality constrained optimization to enforce alignment between local and global update directions. The alignment is done at both the level of the client and server which alleviates non-iidness to a certain extent. Technically speaking, the approach is quite general and original in its form.
+ The experimental results show that the proposed algorithm outperforms other well known baselines.

Weakenesses:
- The alignment at the client level requires the server sending the aggregated model update from the past round to each client, which undermines the privacy-preserving aspect of FL quite a bit. Moreover, the optimization problem to be solved at hand is a constrained optimization with additional computational overhead on the clients, with no guarantees of existence of a feasible space.  It's also not clear how C is chosen and how it affects convergence.
- The discussion in Step 1 in page 4 is erroneous. Unlike what the authors discuss, mini-batch SGD evaluates gradients at the same model iterate for multiple batches and then averages them. It doesn't take update steps after each gradient evaluation as suggested by the authors. Overall, step 1 of the algorithm is no different than standard local mini-batch SGD, where the client takes multiple local steps.
- The question of feasibility of the constrained optimization becomes even more acute in case of the server, as it combines $k$ different constraints. It's not clear to me especially as to what is done, if the intersection of such constraints leads to an empty set, i.e., there's no feasible solution. In sufficiently non-iid data, how does one go about such scenarios.
- The algorithm is devoid of any theoretical guarantees in terms of convergence. It's not clear if the aligned update directions thus obtained lead to a direction of sufficient descent.
- Finally the experimental results leave a lot wanting. Number of local epochs is chosen to be $5$ for the proposed algorithm, while it is taken to be $1$ arbitrarily for all other baselines for the Handwritten digits dataset. Similarly, while learning rate tuning is done for the proposed algorithm, all other baselines are assigned a fixed learning rate without any tuning. Hence, the experimental comparisons are not fair and no claims regarding the superiority of the proposed approach can be made.


**Summary Of The Paper:**

This paper studies the problem of alleviating convergence issues in FL in the face of non-iid data. In particular, the authors propose methods which align the gradient and the direction of update with the past update at the server, and align the server update with the aggregated client model update. The alignment is done through a projection operation. Finally, empirical results on public datasets illustrate the efficacy of the approach.

**Summary Of The Review:**

The paper considers a relevant problem of interest in FL to tackle non-iidness and proposes to tackle it with gradient alignment via projections. There are a lot of questions concerning the algorithm both in terms of feasibility and convergence, which is not covered in the paper. Finally, experimental results are weak in terms of fairness of comparing with other baselines.

---

> ### Author Response · Authors · 2022-11-17
> **Response to Reviewer ESza (Part 1/3)**
>
> Dear Reviewer,
>
> We thank the reviewer for providing helpful feedback to improve our work. We address concerns and questions from the reviewer, and answer them individually.
>
> > **Q1.1: The alignment at the client level requires the server sending the aggregated model update from the past round to each client, which undermines the privacy-preserving aspect of FL quite a bit.**
>
> **Response:** Thanks for the comment. We have provided the privacy-preserving analysis of our FedGC using one popular gradient inversion attack “Deep Leakage from Gradients (DLG)” [1] in Section **Appendix A.5**, and conduct the gradient inversion attacks experiment to compare FedAvg and our FedGC with DLG on CIFAR10-10 dataset. The analysis and experimental result show that our FedGC can ensure the preserving of privacy.
>
> [1] Ligeng Zhu, Zhijian Liu, and Song Han. Deep leakage from gradients. In H. Wallach, H. Larochelle, A. Beygelzimer, F. d'Alch´e-Buc, E. Fox, and R. Garnett (eds.), Advances in Neural Information Processing Systems, volume 32. Curran Associates, Inc., 2019.
>
> > **Q1.2: Moreover, the optimization problem to be solved at hand is a constrained optimization with additional computational overhead on the clients, with no guarantees of existence of a feasible space.**
>
> **Response:** The additional computational overhead of the constrained optimization (i.e., gradient projection) on the clients is actually very low. There are two reasons. First, we adopt the layer-wise manner rather than the whole concatenated gradient to perform gradient projection (in both SGC and CGC), that significantly reduces the memory and computational overhead. For example, ResNet-9 used in our experiments contains $3\times3\times3\times64=1728$ parameters (as the same as gradients) in its first layer and $3\times3\times64\times128=73728$ in its second layer, and so on. The way of concatenating all flattening gradients in ResNet-9 into a gradient vector will cause the dimension of variables in Eqs.(6) and (9) being huge ($p = 1728+ 73728+...$) and dramatically increases the memory space for gradient projection. In contrast, the layer-wise manner in our work (i.e., we iteratively perform the gradient projection layer by layer) is very efficient and does not require a large memory space. Second, at client, we convert the primal problem on $p$ variables of client projection method into a dual problem on $1$ variable. This can greatly reduce the computational complexity and time. Therefore, the constrained optimization in our FedGC does not require a large memory space and it can guarantee that there is a feasible space in most applications.
>
> > **Q1.3: It's also not clear how C is chosen and how it affects convergence.**
>
> **Response:** In our method, $C$ is a small positive constant to avoid $g_k^t$ and $g^{t-1}$ being orthogonal after projection. Following the reviewer suggestions, we conduct the ablation experiment on HWDigits-4 to explore its effect on convergence. The table below shows the experimental result.
>
> | $C$   | $1e-1$ | $1e-2$ | $1e-3$    | $1e-4$ | $1e-5$ |
> | ----- | ------ | ------ | --------- | ------ | ------ |
> | Acc.  | --     | --     | **95.27** | 93.59  | 93.47  |
> | Round | --     | --     | **359**   | 376    | 420    |
>
> If $C$ is a large one (e.g., $C=1e-1$, $C=1e-2$), the value of $<g_k^t, g^{t-1}>$ may be smaller than $C$ during training, and this is meaningless. If $C$ is a small one (e.g., $C=1e-5$), it may slow down the convergence rate. Therefore, for consistency, in our method, $C=1e-3$ for all experiments.

---

> > ### Author Response · Authors · 2022-11-17
> > **Response to Reviewer ESza (Part 2/3)**
> >
> > > **Q2: (1) The discussion in Step 1 in page 4 is erroneous. Unlike what the authors discuss, mini-batch SGD evaluates gradients at the same model iterate for multiple batches and then averages them. It doesn't take update steps after each gradient evaluation as suggested by the authors. (2) Overall, step 1 of the algorithm is no different than standard local mini-batch SGD, where the client takes multiple local steps.**
> >
> > **(1) Response:** We apologize for mis-using symbols, and correct this error in Step 1 of page 4 by replacing the statement
> > "Instead of the mean gradient $\bar{g}_k^t$=
> >
> > $\frac{1}{E}\sum_{l=1}^{E} g^t_{k,l}$ used in Minibatch SGD"
> > with
> > "Different from the mean gradient $\bar{g}_k^t$=
> >
> > $\frac{1}{B}\sum_{b=1}^{B}g^t_{k,b'}$ used in Minibatch SGD that averages multiple gradients for several mini-batches data at the same point (e.g., $\theta_{i,0}^t$)", in the revised manuscript. In addition, **Figure 1** has been revised in the the revised manuscript by adding the calculation of mean gradient $\bar{g}_k^t$ (in blue dot rectangle boxes) in Minibatch SGD, to distinguish it from our pseudo gradient $\tilde{g}_k^t$.
> >
> > **(2) Response:** Here are two differences between step 1 and the standard local mini-batch SGD. First, our FedGC performs ***multiple mini-batches*** to train local model in each round rather than performing ***multiple epochs*** in standard local mini-batch SGD. Second, the pseudo-gradient is not simply $\theta_{k,B}^t-\theta_{k, 0}^t$ but scaled by $1/\eta$, that can promote a large update for local model to accelerate the convergence rate.
> >
> >
> > > **Q3: The question of feasibility of the constrained optimization becomes even more acute in case of the server, as it combines k different constraints. It's not clear to me especially as to what is done, if the intersection of such constraints leads to an empty set, i.e., there's no feasible solution. In sufficiently non-iid data, how does one go about such scenarios.**
> >
> > **Response:** We agree with the reviewer that “there’s no feasible solution” at some time. In Page 6, we have provided the solution by using “$\bar{g}^t$ to be $g^t$ ” when the problem (9) is unsolvable. More studies may be considered as future works.
> >
> > >  **Q4: The algorithm is devoid of any theoretical guarantees in terms of convergence. It's not clear if the aligned update directions thus obtained lead to a direction of sufficient descent.**
> >
> > **Response:** We have provided the theoretical analysis of gradient projection in **Appendix A.3** and convergence analysis in **Appendix A.4**.

---

> > > ### Author Response · Authors · 2022-11-17
> > > **Response to Reviewer ESza (Part 3/3)**
> > >
> > > > **Q5.1: Finally the experimental results leave a lot wanting. Number of local epochs is chosen to be 5 for the proposed algorithm, while it is taken to be 1 arbitrarily for all other baselines for the Handwritten digits dataset.**
> > >
> > > **Response:** We apologize for confusing the reviewer that we use a symbol $E$ to represent different definitions in FedGC (our) and other compared methods in the original manuscript.
> > >
> > > Our FedGC adopts ***multiple mini-batches*** to train local models at each communication round, while other compared methods train their local models by performing ***multiple epochs*** on full local data. For clarity, we define $B$ as the number of mini-batches used in our FedGC and define $E$ as the number of epochs in other compared methods. In experiments on Handwritten digits datasets, $E=1$ for all other compared methods and $B=50$ for our FedGC.
> > >
> > > Under batch size 100, for HWDigits-4, there are about $73\sim732$ mini-batches per client in $1$ epoch. Hence, under the setting that $E=1$ for other methods and $B=50$ for our FedGC, the local model in other methods performs more iterations than our FedGC in each communication round. Therefore, we did not unfairly set the number of epochs for compared methods.
> > >
> > > > **Q5.2: Similarly, while learning rate tuning is done for the proposed algorithm, all other baselines are assigned a fixed learning rate without any tuning. Hence, the experimental comparisons are not fair and no claims regarding the superiority of the proposed approach can be made.**
> > >
> > > **Response:** Due to space limitation, some details of tuning learning rate were missing in the original manuscript. In fact, the learning rates $\eta$ of all compared methods are tuned in experiments. The learning rates of all methods in experiment are tuned in range {$0.01, 0.05, 0.1, 1.0$}. The test accuracies of all compared methods on HWDigits-4 are shown below. Hence, $\eta=0.01$ and $\eta=0.1$ are the optimal ones for all other compared methods and our FedGC respectively, which is consistent with the setting in 4.2. Hence, the experimental comparisons are fair and we have provided both theoretical and empirical verifications in the revised manuscript.
> > >
> > >
> > >
> > > | $\eta$   | 0.01  | 0.05  | 0.1   | 1.0     |
> > > | -------- | ----- | ----- | ----- | ------- |
> > > | FedAvg   | 93.07 | 92.72 | 91.63 | NaN$^1$ |
> > > | FedProx  | 93.69 | 92.93 | NaN   | NaN     |
> > > | FedCurv  | 91.78 | 91.97 | NaN   | NaN     |
> > > | SCAFFOLD | 92.29 | 92.08 | NaN   | NaN     |
> > > | FedReg   | 92.32 | 92.04 | NaN   | NaN     |
> > > | FedGC    | 95.03 | 94.64 | 95.27 | NaN     |
> > > $^1$ large learning rates result in gradient explosion.

---

> ### Author Response · Authors · 2022-12-11
> **Looking forward to your feedback.**
>
> Dear Reviewer ESza,
>
> Thanks again for providing a valuable pre-rebuttal review! We have responded to your initial comments. We are looking forward to your feedback and would be happy to address any further concerns you may have.
>
> Thank you, author

---

### Author Response · Authors · 2022-11-17
**Thank you all for your comments and feedback**

We would like to thank the reviewers for providing high-quality reviews and constructive feedback that have improved the paper. We are encouraged that the reviewers think our paper is “Technically speaking, the approach is quite general and original in its form.” (Reviewer **ESza**), “the proposed methods are original” (Reviewer **NKr4**), “The proposed methods are intuitive, easy to understand.” (Reviewer **yV7y**).

We found the constructive feedbacks from the reviewers very helpful and have prepared an updated version of our manuscript. Revisions to the manuscript are marked in red in the revised manuscript and summarized below. The appendix is also extended with theoretical analysis and more required experiments. We reply to each reviewer in more detail in individual responses.
- Following the suggestions of reviewers **ESza** and **yV7y**, we have provided the theoretical analysis of gradient projection in Appendix A.3 and convergence analysis of our FedGC in A.4.
- Following the suggestions of reviewers **ESza** and **NKr4**, we have provided the privacy-preserving analysis of our FedGC using one popular gradient inversion attack “Deep Leakage from Gradients (DLG)” in Appendix A.5.
- We have corrected the erroneous discussion about Minibatch SGD raised by reviewers **ESza** and **NKr4** in the revised manuscript.
- We have improved the clarity of the experimental setting to avoid confusing reviewers **ESza** and **NKr4**, e.g., our FedGC adopts multiple mini-batches rather than multiple epochs to train local models, and the learning rate tuning.
- Based on the comments of reviewer **NKr4**, we have revised the pseudo code of our FedGC and confirmed that our FedGC only transfers gradients between server and clients.
- Based on the comments of reviewer **yV7y**, we have provided more illustrations of "catastrophic forgetting" in the introduction and given an example in Appendix A.10 to showcase that the forgetting issue is important in FL.
- Based on the comments of reviewer **yV7y**, we have given more comparisons to show the novelty of the proposed method.

---

### Decision · Program_Chairs · 2023-01-20

**Decision:**

Reject

**Justification For Why Not Higher Score:**

Several ways in which the paper could be improved (clarifying algorithm description, clearer experimental details, more thorough experiments).

**Justification For Why Not Lower Score:**

N/A

**Metareview: Summary, Strengths And Weaknesses:**

This paper introduces an approach to address heterogeneity in federated learning (FL) by constraining client update directions to be well-aligned. The potential of the proposed approach is demonstrated via experiments on a dataset of handwritten digits (combination of MNIST, MNIST-Z, USPS, and SVHN), and CIFAR-10/100.

The initial experiments illustrate that the approach may be promising.

However several weaknesses were noted, including:
* The proposed approach may not be practical for FL implementations or may be incompatible with privacy requirements (requiring each device's gradient to be exposed to the server could enable reconstruction attacks)
* Experiments have several limitations (several details about experimental setup and comparison to previous work were not clear from the submission)
* Some aspects of the algorithm description were not clear from the submission.

**Summary Of Ac-Reviewer Meeting:**

n/a